# Action-Sufficient Goal Representations

**Jinu Hyeon** [* 1]   **Woobin Park** [* 2]   **Hongjoon Ahn** [* 2]   **Taesup Moon** [1 2 3]

## Abstract

In offline goal-conditioned reinforcement learning (GCRL), hierarchical approaches decompose long-horizon tasks into high-level subgoal prediction and low-level action execution. A critical design choice in such architectures is the goal representation—the compressed encoding of goals that serves as the interface between these levels. Existing methods derive this representation from value learning, implicitly assuming that information sufficient for value estimation is adequate for optimal action prediction. We show that this assumption can fail even under exact value estimation, as such representations may collapse goals requiring distinct optimal actions. To address this, we introduce *action sufficiency*, an information-theoretic condition on goal representations necessary for optimal action prediction. We prove that value sufficiency, the preservation of sufficient information for value estimation, does not imply action sufficiency and empirically verify that the latter is more strongly associated with control success in a discrete environment. We further demonstrate that an actor-based representation, naturally induced by standard log-likelihood training of the low-level policy, is approximately action-sufficient. Empirically, our actor-based representations consistently outperform representations learned via value function estimation.

## 1. Introduction

The primary advantage of offline goal-conditioned reinforcement learning (GCRL) lies in its ability to extract diverse and complex skills from pre-collected datasets without further environment interaction (Park et al., 2025b; Chane-Sane et al., 2021; Eysenbach et al., 2022; Liu et al., 2022; Ma et al., 2022). However, the efficacy of this paradigm fundamentally depends on how state and goal information are encoded into a shared representation space to bridge the gap between observed data and target behaviors.

Goal representations become even more critical in long-horizon tasks, where hierarchical frameworks are often employed. In such systems, a high-level policy provides subgoals to a low-level policy, and the effectiveness of this hierarchy hinges on the quality of the goal representation space used to communicate these subgoals. A prominent example of this approach is HIQL (Park et al., 2023), which learns goal representations directly through the value learning objective and utilizes them to facilitate subgoal prediction. While HIQL has demonstrated significant empirical success in complex, long-horizon environments, a systematic understanding of why value-centric representations work—and specifically how they relate to the downstream low-level policy—remains incomplete. Park et al. (2025d) recently provided a rigorous interpretation from the standpoint of value estimation, but we note a critical gap persists: the lack of a policy-oriented analytical framework.

Our work challenges the implicit assumption of previous approaches that features optimized for learning value functions naturally facilitate policy learning. To this end, we first conduct empirical evaluations on the `cube` task from OGBench (Park et al., 2025a), specifically measuring the performance of low-level policies. Our results reveal a significant failure mode: agents utilizing value-centric representations exhibit markedly low success rates, suggesting that such features may lack critical information for effective action selection.

To explain this phenomenon, we introduce the formal concepts of action sufficiency and value sufficiency, defined by whether a goal representation preserves necessary information for accurate optimal policy derivation or value prediction, respectively. Using these definitions, we theoretically demonstrate that value sufficiency does not inherently guarantee action sufficiency of goal representations. We show that representations optimized solely to reflect value function can discard relative action-relevant nuances, leading to a fundamental disconnect between value estimation accuracy and policy success. This theoretical finding is

---
\* Equal Contribution [1]Interdisciplinary Program in Artificial Intelligence (IPAI), Seoul National University [2]Department of Electrical and Computer Engineering (ECE), Seoul National University [3]ASRI / INMC, Seoul National University. Correspondence to: Taesup Moon <tsmoon@snu.ac.kr>.

*Proceedings of the 43rd International Conference on Machine Learning*, Seoul, South Korea. PMLR 306, 2026. Copyright 2026 by the author(s).

further validated in a discrete environment, where control success aligns with action sufficiency rather than value sufficiency, even among representations that preserve the optimal value.

Consequently, we propose a simple approach that integrates representation learning directly into the actor's objective, encouraging the representation to preserve the information-theoretic requirements for optimal action prediction. On the OGBench benchmark, we verify that replacing value-based representations with our actor-based ones yields consistent improvements, particularly on high-difficulty tasks where value-based representations collapse.

## 2. Related Work

**Offline goal-conditioned reinforcement learning.** Offline GCRL learns goal-reaching policies from pre-collected datasets without further environment interaction, combining the dataset constraints and distribution-shift issues of offline RL (Levine et al., 2020) with the goal-reaching objectives of GCRL (Schaul et al., 2015; Andrychowicz et al., 2017; Ren et al., 2019; Liu et al., 2022; Zheng et al., 2024).

One of the primary approaches to tackling offline GCRL, which constitutes the main focus of our analysis, is to extract a policy from a learned value function. These methods exploit datasets by learning goal-conditioned value functions via temporal-difference (TD) learning (Kostrikov et al., 2022; Park et al., 2024), state-occupancy matching (Ma et al., 2022; Durugkar et al., 2021), contrastive representations (Ma et al., 2023; Eysenbach et al., 2022; Liu et al., 2025), or quasimetrics (Wang et al., 2023; Myers et al., 2025a). Another approach involves diffusion-based planners (Janner et al., 2022; Luo et al., 2025), which generate trajectories directly without relying on value-derived policies, placing them outside the scope of our analysis.

**Hierarchical methods for GCRL.** Long-horizon GCRL is typically addressed by decomposing decision-making into a high-level subgoal planner and a low-level controller (Dayan & Hinton, 1992; Kulkarni et al., 2016; Vezhnevets et al., 2017; Nachum et al., 2018; Pateria et al., 2021), a paradigm instantiated in the offline setting by HIQL (Park et al., 2023). Recent advances focus on improving value estimation within this regime via temporal abstraction (Ahn et al., 2025; Park et al., 2025b) or physics-informed regularization (Giammarino et al., 2025).

**Goal representations in GCRL.** How goals are represented critically affects policy learning and generalization in GCRL. A variety of representation learning methods have been studied, including embeddings that encode temporal distances (Sermanet et al., 2018; Ma et al., 2023; Park et al., 2024; 2025d) and trajectory-level contrastive representations (Eysenbach et al., 2022; Myers et al., 2025b; Lawson

et al., 2026). In the hierarchical setting of offline GCRL, a goal representation serves as an interface between high- and low-level policies, and is conventionally learned from value-related signals. Specifically, HIQL (Park et al., 2023) theoretically established the sufficiency of a goal-only encoder $\phi(g)$ trained via value optimization, but practically adopted a state-dependent representation $\phi(s, g)$ following (Hong et al., 2022). While this state-dependent design is both mathematically and empirically intuitive, a formal justification for its efficacy has been lacking. We provide this missing rationale from an action-oriented standpoint.

**Action-aware representations in RL.** Outside offline GCRL, the principle that representations should retain action-relevant information has been pursued through policy-derived representations (Ghosh et al., 2019; Garcin et al., 2025), inverse-dynamics objectives (Lamb et al., 2023; Islam et al., 2023; Brandfonbrener et al., 2023), structural formulations of action-sufficient state (Huang et al., 2022), and similarity metrics that capture action-relevant structure (Agarwal et al., 2021; Rudolph et al., 2024). While these works establish the necessity of action-aware representations in single-policy settings, the hierarchical offline GCRL framework remains unexamined. Our contribution lies in providing the first information-theoretic formalization and empirical verification of action sufficiency specifically as a required property for goal representations serving as a hierarchical interface.

## 3. Preliminary

**Problem setting.** GCRL is formulated as a Markov decision process (MDP) defined by the tuple $(\mathcal{S}, \mathcal{A}, \mathcal{G}, r, \gamma, p, p_0, p_{\mathcal{G}})$. Here, $\mathcal{S}$ denotes the state space, $\mathcal{A}$ the action space, and $\mathcal{G}$ the goal space. The reward function $r(s, g)$ is conditioned on both the current state $s \in \mathcal{S}$ and the goal $g \in \mathcal{G}$, $\gamma \in (0, 1)$ is the discount factor, $p(\cdot \mid s, a)$ is the environment transition dynamics, $p_0(\cdot)$ is the initial state distribution, and $p_{\mathcal{G}}$ denotes the goal distribution. We denote by $V(s, g)$ the goal-conditioned state-value function at state $s$ with respect to goal $g$. Throughout this work, we assume that the goal space coincides with the state space, *i.e.*, $\mathcal{G} = \mathcal{S}$.

In the offline setting, the agent is given access to a fixed dataset $\mathcal{D}$ consisting of trajectories $\tau = (s_0, a_0, s_1, \ldots, s_T)$, collected by an unknown behavior policy $\mu$. The objective is to learn a goal-conditioned policy $\pi(a \mid s, g)$ that maximizes the expected discounted return

$$\mathcal{J}(\pi) = \mathbb{E}_{\tau \sim p^\pi, \, g \sim p_{\mathcal{G}}} \left[ \sum_{t=0}^{T} \gamma^t r(s_t, g) \right], \qquad (1)$$

in which the trajectory distribution induced by $\pi$ is given by $p^\pi(\tau) = p_0(s_0) \prod_{t=0}^{T-1} p(s_{t+1}|s_t, a_t) \pi(a_t|s_t, g)$. For

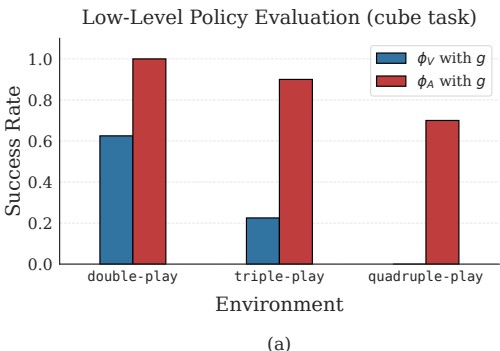
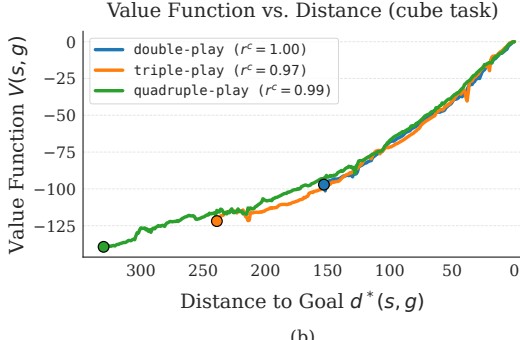

(a)

(b)

*Figure 1.* *Left*: The results of low-level policy evaluation on short-horizon subgoal $s_{t+k}$ from optimal trajectory $\tau^\star$. *Right*: The visualization of value function $V(s, g)$ by varying $s$ from initial state to goal $g$ in $\tau^\star$. Although the value function effectively captures the discounted temporal distance throughout all environments and thereby achieves high order consistency, the low-level policy equipped with $\phi_V$ attains much lower success rate than the counterpart which uses $\phi_A$.

our analysis, we further assume a fixed optimal policy $\pi^\star$, ensuring that the optimal action distribution is well-defined given the current state $s$ and goal $g$.

**Hierarchical policies in GCRL.** To solve long-horizon goal-reaching tasks, hierarchical approaches decompose decision-making into two levels: a high-level policy (the *planner*) and a low-level policy (the *controller*). A representative framework is HIQL (Park et al., 2023), which we adopt as our main analysis target. The high-level policy $\pi^h(s_{t+k} \mid s_t, g)$ proposes a $k$-step subgoal to guide progress toward a distant goal $g$, while the low-level policy $\pi^\ell(a_t \mid s_t, s_{t+k})$ controls the agent through primitive actions to reach the subgoal. Both policies are trained via advantage-weighted regression (AWR) (Peng et al., 2019), which maximizes

$$\mathcal{J}(\pi^i) = \mathbb{E}_\mathcal{D}\left[ e^{\beta^i A^i} \log \pi^i \right], \quad i \in \{h, \ell\}, \quad (2)$$

where the advantages $A^h = V(s_{t+k}, g) - V(s_t, g)$ and $A^\ell = V(s_{t+1}, s_{t+k}) - V(s_t, s_{t+k})$ weight the behavioral cloning term toward high-value transitions and $\beta^i$ are inverse temperatures. We defer how the underlying value function is learned to Appendix A.

**Goal representation learning.** A critical component of the hierarchical framework is how subgoals are represented and communicated between the policies. In practice, HIQL learns a *value-based goal representation* $\phi_V(s, g)$ as part of the value function training process: the value function is of the form $V(s, \phi_V(s, g))$, where the representation $\phi_V$ is obtained by concatenating state and goal features and is optimized to minimize the value estimation error. Once learned, this same representation $\phi_V$ is frozen and used as a proxy for the subgoal $s_{t+k}$ when training both policies, *i.e.*, $\pi^h(\phi_V(s_t, s_{t+k}) \mid s_t, g)$ and $\pi^\ell(a_t \mid s_t, \phi_V(s_t, s_{t+k}))$. Thus the goal representation $\phi_V$, learned solely from value estimation, serves as the interface between the policies.

## 4. Motivation

### 4.1. Are Value-based Goal Representations Sufficient for action prediction?

The value-based representation $\phi_V$ described above implicitly assumes that features optimized for **predicting values** are sufficient for **action prediction** (Park et al., 2023; 2025d). In this section, we investigate this assumption through an experiment inspired by Ahn et al. (2025), designed to isolate the controllability of the low-level policy.

**Oracle subgoal evaluation.** To isolate low-level controllability from high-level planning errors, we replace predicted subgoals with the oracle planner provided by OGBench (Park et al., 2025a). Specifically, given an optimal trajectory $\tau^\star = (s_0, s_1, \cdots, s_T)$, the policy at state $s_t$ conditions on the exact oracle subgoal $s_{t+k}$. In this setting, we train low-level policies using two different goal represetations: the original HIQL **value-based goal representation** $\phi_V(s, g)$ learned by optimizing the value objective against the **actor-based goal representation** $\phi_A(s, g)$, which is learned jointly with the low-level policy $\pi^\ell(a_t \mid s_t, \phi_A(s_t, g))$ via optimizing the AWR loss in Eq. (2). Note that both low-level policies are trained to reach the original goal $g$ rather than the subgoal $s_{t+k}$, encouraging the representation to generalize across the goal space. Since the subgoal and the final goal lie in the same space ($\mathcal{G} = \mathcal{S}$), the low-level policy can be directly conditioned on the oracle subgoal $s_{t+k}$ from the optimal trajectory $\tau^\star$ during evaluation.

**The discrepancy between value and control.** Under this evaluation protocol, we measure the goal success rates of both low-level policies across the OGBench `cube` tasks. As shown in Figure 1(a), the policy using $\phi_A$ significantly outperforms the policy using $\phi_V$, despite both receiving perfect subgoal information from the oracle. This performance gap indicates that $\phi_V$ can fail to capture information necessary for action prediction.

**Is the value function reliable?** One might hypothesize that this failure stems from poor training of the value function itself rather than an inherent limitation of the representation. To investigate this, we measure the *order consistency ratio* $r^c(V)$, a metric proposed by Ahn et al. (2025) to evaluate the reliability of learned value functions. This ratio is defined as the proportion of transitions along an optimal trajectory $\tau^\star$ for which the value correctly increases toward the goal:

$$r^c(V) = \frac{1}{T-k+1} \sum_{t=0}^{T-k} \mathbf{1}\left[V(s_{t+k}, g) > V(s_t, g)\right].$$

As illustrated in Figure 1(b), the trained value exhibits high order consistency ratio, confirming that the value network successfully learned the temporal structure of the task.

**Summary.** These findings reveal that representations optimized for value estimation are not optimal for action prediction, even when the value estimates themselves are reliable. This discrepancy suggests that the information required to *evaluate* a state differs from that required to *act* upon it to reach the goal.

### 4.2. Value Prediction $\neq$ Action Prediction: A 1D Thought Experiment

To illustrate why a value-preserving goal encoder can be structurally ill-suited for action selection, we analyze a minimal thought experiment on the integer line, adopting the setting from the motivating example in Park et al. (2023).

**Setup.** Consider a deterministic MDP where $\mathcal{S} = \mathcal{G} = \mathbb{Z}$ and actions $\mathcal{A} = \{+1, -1\}$ shift the state by $a$. Under the shortest-path reward $r(s, g) = -\mathbf{1}[s \neq g]$, the optimal value depends purely on the distance:

$$V^\star(s, g) = -\frac{1 - \gamma^{|s-g|}}{1 - \gamma}.$$

**Representations.** We compare two distinct goal representations :

$$\phi_{\text{sign}}(s, g) = s - g, \qquad \text{(Preserves direction)}$$
$$\phi_{\text{dist}}(s, g) = |s - g|. \qquad \text{(Preserves distance only)}$$

We defer the justification for allowing $\phi$ to depend on $(s, g)$ rather than solely on $g$ to Appendix B. Note that both representations allow recovering $V^\star$ exactly (*i.e.*, $V^\star$ is a function of $|s - g|$).

**The conflict.** While $\phi_{\text{dist}}$ is perfect for value prediction, it fails for control. Consider a state $s$ and two opposite goals:

$$g^+ = s + 1 \implies a^\star = +1.$$
$$g^- = s - 1 \implies a^\star = -1.$$

Here, $\phi_{\text{dist}}$ maps both goals to the same embedding: $\phi_{\text{dist}}(s, g^+) = \phi_{\text{dist}}(s, g^-) = 1$. Consequently, a policy

conditioning on $\phi_{\text{dist}}$ receives identical inputs for distinct optimal actions, making optimal control impossible. In contrast, $\phi_{\text{sign}}$ distinguishes these cases ($-1$ vs $+1$), enabling the policy to solve the task.

**Key message.** *A representation can retain all information needed to recover the optimal value while still collapsing distinct goals that require different optimal actions.* This motivates analyzing goal representations through the lens of action-oriented perspective, rather than a value-centric one.

## 5. Optimal Policy Estimation and Action Sufficiency

The previous counterexample motivates a fundamental question: *what properties should a goal representation satisfy to support optimal action selection?* We formalize this via the **conditional KL risk**, measuring the divergence between the optimal policy and a learned predictor restricted to the representation. This risk decomposes into modeling error and representation error, motivating our core definition: a representation is *action-sufficient* if the representation error vanishes, guaranteeing that compression imposes no ceiling on performance.

### 5.1. Probabilistic Setup

Following Section 3, we treat the state, goal, and optimal action as random variables $S$, $G$, and $A$, taking values in measurable spaces $\mathcal{S}$, $\mathcal{G}$, and $\mathcal{A}$, respectively. For notational convenience, we drop the superscript $\star$ from the optimal action $A^\star$. Their joint law $P$ is fully specified, as the joint distribution of $(S, G)$ is fixed and the conditional distribution of the optimal action given $(S, G)$ is assumed to be well-defined. We consider a deterministic representation $Z := \phi(S, G)$ for a measurable function $\phi$.

We frame control within a hierarchical architecture. The *high-level policy* $\pi^h$ proposes a stochastic subgoal $G_{\text{sub}} \in \mathcal{G}$ given $(S, G)$, inducing a distribution over representations $Z_{\text{sub}} := \phi(S, G_{\text{sub}})$:

$$\pi^h(\cdot \mid S, G) \in \mathcal{P}(\mathcal{Z}).$$

The *low-level policy* $\pi^\ell$ selects actions conditioned on the state and subgoal representation:

$$\pi^\ell(\cdot \mid S, Z_{\text{sub}}) \in \mathcal{P}(\mathcal{A}).$$

At deployment, the low-level policy conditions on representations predicted by the high-level policy. Since subgoals and final goals share the same space, we use the notation interchangeably hereafter.

### 5.2. Conditional KL Decomposition

We analyze the predictability of the low-level policy $\pi^\ell$ induced by the representation $\phi$. To measure the discrepancy

between the optimal policy $P(A|S,G)$ and its predictor, we employ the **conditional KL risk**:

$$\mathcal{R}(\pi^\ell; \phi) = \mathbb{E}\left[D_{\mathrm{KL}}(P(A|S,G) \,\|\, \pi^\ell(A|S,Z))\right]. \quad (3)$$

Our analysis yields the following decomposition of this risk:

> **Proposition 5.1** (Conditional KL Risk Decomposition). *The risk $\mathcal{R}(\pi^\ell; \phi)$ decomposes into:*
>
> $$\mathcal{R}(\pi^\ell; \phi) = \underbrace{\mathbb{E}\left[D_{KL}(P(A|S,Z) \,\|\, \pi^\ell(A|S,Z))\right]}_{\textit{Modeling Error}}$$
> $$+ \underbrace{I(A; G \mid S, Z)}_{\textit{Representation Error}},$$
>
> *where $I(\cdot; \cdot | \cdot)$ denotes conditional mutual information.*

Full proofs and details are deferred to Appendix C.

### 5.3. Action Sufficiency

Proposition 5.1 reveals that optimizing $\pi^\ell$ to minimize Eq. (3) serves only to minimize the modeling error, leaving the second term as an irreducible penalty determined solely by the information loss in $\phi$. This motivates our core definition of **action sufficiency**:

> **Definition 5.2** (Action-Sufficient Representation). A representation $Z = \phi(S,G)$ is *action-sufficient* if:
>
> $$\Delta_A := I(A; G \mid S, Z) = 0.$$

This condition is equivalent to the conditional independence $A \perp G \mid (S, Z)$, or almost surely matching the posterior laws: $P(\cdot|S,G) = P(\cdot|S,Z)$. Intuitively, action sufficiency implies that $Z$ retains all goal information requisite for optimal action prediction.

This definition is not arbitrary; it arises from the fundamental conditional KL gap incurred when approximating the goal-conditioned action posterior $P(\cdot \mid S, G)$ using a policy restricted to conditioning on $(S, Z)$, with $\Delta_A$ capturing the resulting irreducible residual. If $\Delta_A > 0$, no policy $\pi^\ell(\cdot \mid S, Z)$ can drive the risk in Eq. (3) to zero, even with infinite data and perfect optimization.

## 6. Value Sufficient Representation Is Not Action Sufficient

In the previous section, we established the concept of **action sufficiency** ($\Delta_A = 0$). Here, we investigate why representations trained solely to predict values, which is a common practice in hierarchical methods for GCRL (Park et al., 2023; Ahn et al., 2025; Park et al., 2025d), often fail to achieve this standard. We provide detailed proofs for the results presented in this section in Appendix D.

### 6.1. Optimal Goal-Conditioned Value Function and Value Sufficiency

First, we formalize the value-based perspective. With a discount factor $\gamma \in (0, 1]$ and a goal-conditioned reward function $r(s, g)$, the *optimal goal-conditioned value function* is defined as:

$$V^\star(s, g) = \sup_\pi \mathbb{E}\left[\sum_{t \geq 0} \gamma^t r(S_t, g) \,\middle|\, S_0 = s\right]$$

where the supremum is taken over all possible policies. Following the notation convention established in the definition of action sufficiency, we drop the superscript $\star$ and treat the value as a random variable $V := V(S, G)$.

Analogous to action sufficiency, we define the value sufficiency of a representation $\phi$ as follows:

> **Definition 6.1** (Value-Sufficient Representation). A representation $Z = \phi(S, G)$ is said to be *value-sufficient* if:
>
> $$\Delta_V := I(V; G \mid S, Z) = 0 \quad (4)$$

Since $V$ is deterministic with respect to $(S, G)$, value sufficiency is equivalent to stating that $V$ can be expressed as a function of $(S, Z)$, *i.e.*, there exists a measurable $\tilde{v}$ such that $V = \tilde{v}(S, Z)$.

### 6.2. Value Sufficiency Does Not Guarantee Action Sufficiency

While value sufficiency ensures accurate value prediction, it does not necessarily guarantee action sufficiency ($\Delta_A = 0$). The following proposition provides an exact decomposition of the action sufficiency gap when a representation is value-sufficient.

> **Proposition 6.2** (Action Information Decomposition). *If a representation $Z = \phi(S, G)$ is value-sufficient (i.e., $\Delta_V = 0$), then the action sufficiency gap $\Delta_A$ decomposes as:*
>
> $$\Delta_A = I(A; G \mid S, V) - I(A; Z \mid S, V) \quad (5)$$

This proposition reveals a critical insight: even if $Z$ preserves the optimal value perfectly, it can still be action-insufficient unless it retains *action-discriminative information among goals that share the same value*.

Conditioning on the value $V$ partitions the goal space into *value level sets*, $\{g : V^\star(s, g) = v\}$. Within a fixed value level set, goals are indistinguishable in terms of the optimal value, yet can require different optimal actions (*e.g.*, goals $N$ steps away in opposite directions). This mismatch is captured by the residual action-relevant goal information

$I(A; G \mid S, V)$, *i.e.*, the action uncertainty that remains even when the scalar value is known.

The extent to which a representation $Z$ resolves this *within-level-set ambiguity* is quantified by $I(A; Z \mid S, V)$: it measures how much of the residual uncertainty $I(A; G \mid S, V)$ is explained by $Z$ beyond the value signal. At the extreme, if $I(A; Z \mid S, V) = 0$, then the action gap reduces to

$$\Delta_A = I(A; G \mid S, V),$$

meaning that perfect value sufficiency yields no reduction in action-relevant goal information beyond what is already captured by $V$.

This phenomenon explains the failure observed in the 1D example in Section 4. In that example, there existed distinct goals $g^+$ and $g^-$ that shared the same optimal value but required different optimal actions. A representation that failed to distinguish these goals (*e.g.*, $\phi_{\text{dist}}$) resulted in $I(A; Z \mid S, V) \approx 0$, creating a structural indistinguishability that prevented optimal control.

More generally, in many non-trivial environments, $I(A; G \mid S, V)$ is strictly positive. Conditioning on the optimal value alone typically does not resolve goal-dependent action ambiguity. We defer a concrete and plausible sufficient condition guaranteeing $I(A; G \mid S, V) > 0$ to Appendix E.

In the subsequent section, we empirically demonstrate that even in discrete environments, there exist near-worst-case scenarios where a representation achieves near-perfect value reconstruction ($\Delta_V \approx 0$) while capturing almost no distinct action information ($I(A; Z \mid S, V) \approx 0$). This leads to an irreducible control gap where $\Delta_A \approx I(A; G \mid S, V) > 0$.

---

**Takeaway**

We propose *action sufficiency* ($I(A; G \mid S, Z) = 0$) as a critical prerequisite for optimal control. We prove that *value sufficiency does not imply action sufficiency*: a representation can perfectly predict values yet fail to predict actions.

---

# 7. Discrete Cube: Exact Information-Theoretic Analysis

The preceding theoretical analysis established action sufficiency as a fundamental requirement for low-level control. We specifically highlighted that for value-sufficient representations, the conditional mutual information $I(A; Z \mid S, V)$ emerges as the key quantity governing action sufficiency, representing the information necessary to resolve goal-dependent action ambiguity that persists within value level sets. To empirically validate these findings, we now turn to a controlled environment that admits exact information-theoretic computation.

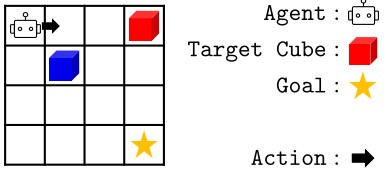

*Figure 2.* **The Discrete Cube envornment.** This domain mirrors the structure of pick-and-place tasks while allowing for the exact computation of information-theoretic quantities. We use this tractability to rigorously verify that control success depends on preserving action sufficiency rather than value sufficiency.

## 7.1. Environment Setup: The Discrete Cube

We construct the *Discrete Cube* (Figure 2), a $4 \times 4$ grid environment that captures the structural essence of goal-conditioned manipulation. The domain features an agent and two distinct cubes (Red and Blue). The state $s$ consists of the agent's coordinates, cube positions, and the gripper status (empty or holding a specific cube). The action space $\mathcal{A}$ comprises four directional movements and two manipulation primitives (pick, place).

A goal $g$ specifies a target cube and its desired spatial coordinate. Success is achieved when the target cube reaches the goal position and is released. Crucially, the finite nature of this domain allows us to compute the *exact* optimal value function $V^\star(s, g)$ and the optimal policy $P^\star(a|s, g)$ via breath-first search (BFS) and derive precise conditional mutual information quantities without estimation error.

## 7.2. Metrics and Control Evaluation

For a fixed representation $Z = \phi(S, G)$, we compute a set of evaluation quantities using the exact environment dynamics: the sufficiency gaps, the within-level-set disambiguation term, and the control success rate.

**Sufficiency gaps.** We quantify divergence from sufficiency by the conditional mutual informations $\Delta_A = I(A; G \mid S, Z)$ for action prediction and $\Delta_V = I(V; G \mid S, Z)$ for value prediction.

**Within-level-set disambiguation.** We measure how much $Z$ helps distinguish optimal actions among state–goal pairs with identical values via $I(A; Z \mid S, V)$.

**Control success rate.** For each goal representation $\phi$, we evaluate the success rate of the mixed policy $\pi_\phi(a \mid s, z) = \mathbb{E}[P^\star(a \mid s, G) \mid S = s, Z = z]$, which acts by marginalizing the optimal action distribution over the posterior goal distribution induced by $Z$.

To systematically investigate the interplay between these metrics, we construct a diverse library of goal representations and randomly sample $\phi$ from this library. We provide full details on this section, including how we construct the representation library, in Appendix F.

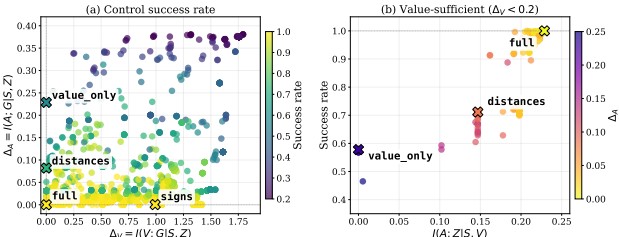

*Figure 3.* **Discrete Cube Result. (a)** Control success rate plotted over the $(\Delta_V, \Delta_A)$ plane for representations $\phi$, evaluated using the mixed policy $\pi_\phi(a \mid s, z)$. **(b)** For (near-) value-sufficient representations ($\Delta_V < 0.2$), control success rate plotted against $I(A; Z \mid S, V)$, with points colored by $\Delta_A$.

### 7.3. Results

To facilitate interpretation of the results, we first introduce four representative baselines that anchor the extremes of action and value information: (1) `full`, which preserves the full relative-position information and is both action- and value-sufficient by construction; (2) `signs`, which keeps only the signs of the displacement components (directional information) while discarding magnitudes; (3) `value_only`, which collapses $Z$ to the scalar $V^\star$, making it perfectly value-sufficient yet highly action-insufficient; and (4) `distances`, which encodes per-coordinate goal distances, remaining value-sufficient while discarding the directional information required to select actions.

$\Delta_A$ **and** $\Delta_V$ **are not structurally coupled.** Figure 3(a) places each representation in the $(\Delta_V, \Delta_A)$ plane and colors points by the success rate under $\pi_\phi$. Before turning to performance, the result illustrates that the two information losses $\Delta_A$ and $\Delta_V$ capture different aspects and need not move together. Importantly, we are *not* asserting that $\Delta_A$ and $\Delta_V$ are uncorrelated as random variables: when the representation $\phi$ is treated as random, $\Delta_A(\phi)$ and $\Delta_V(\phi)$ are induced random variables, and their distributions highly depend on the prior over $\phi$ (here, the representation library and sampling procedure). Since any finite library cannot cover the distribution over all possible representation functions, our goal is not to characterize these distributions in general. Rather, our claim is *structural*: driving $\Delta_V$ down does not, in general, force $\Delta_A$ down. A representation may preserve action-relevant structure while discarding value-relevant details (small $\Delta_A$ but large $\Delta_V$), or preserve value-determining signals while collapsing action-distinguishing information (small $\Delta_V$ but large $\Delta_A$). This experiment makes this decoupling explicit by exhibiting representations that realize each regime.

**Success aligns with the action-sufficiency gap.** Across the representation library, success correlates much more strongly with $\Delta_A$ than with $\Delta_V$: representations that retain action-relevant goal information (small $\Delta_A$) tend to

succeed even when they discard value-relevant details, as illustrated by `signs`. Conversely, value-preserving representations can still fail when they collapse action distinctions: `value_only` retains $V^\star$ but cannot disambiguate goals that share the same value, while `distances` improves by preserving goal distances yet remains limited without directional cues. In contrast, `full` succeeds by preserving both.

**For value-sufficient $\phi$,** $I(A; Z \mid S, V)$ **captures the remaining failure mode.** Figure 3(b) focuses on representations that are (near-) value-sufficient ($\Delta_V < 0.2$) and plots success rates against $I(A; Z \mid S, V)$, which measures how much $Z$ disambiguates optimal actions within $(S, V)$ level sets. Within this approximately value-sufficient regime, the remaining performance differences are explained by within-level-set action ambiguity, consistent with the trends observed for the baselines discussed above.

Overall, Discrete Cube provides an exact demonstration that value preservation alone does not guarantee successful control: success under the mixed policy $\pi_\phi$ tracks the action gap $\Delta_A$, and among (near-) value-sufficient representations it is further governed by the within-level-set disambiguation term $I(A; Z \mid S, V)$.

## 8. Method: Actor-Based Goal Representations and Approximate Action Sufficiency

The preceding sections demonstrate that value-sufficient representations can fail for low-level control by discarding action-relevant information within value level sets. This motivates learning a goal representation that is directly aligned with action prediction.

### 8.1. Training vs. Deployment in Hierarchical Control

A key subtlety in hierarchical policy architecture is that the context available to the low-level policy differs between training and deployment. In a deployed hierarchical policy, the low-level policy is not assumed to have direct access to the ground-truth subgoal it must pursue. This is because, in many tasks, the specific intermediate subgoal (or its ground-truth representation) is not provided externally; instead, the low-level policy acts only on the current state and the subgoal representation predicted by the high-level policy.

During training, however, the availability of the full trajectory allows us to derive the target subgoal $G_{\text{sub}}$. Consequently, the low-level policy can condition on $G_{\text{sub}}$ (or effectively $G$, as subgoals reside in the same space as final goals) during the learning phase. This discrepancy is crucial: *It allows us to learn a goal encoder from the full goal $G$ during training and use the same representation space to condition the low-level policy on subgoals at deployment.*

## 8.2. Actor-Based Goal Representations

We capture the training-time access to $G$ by structuring the full architecture of the low-level policy as:

$$\pi_\theta(\cdot \mid S, Z), \qquad Z = \phi(S, G), \tag{6}$$

where $\phi$ is a goal encoder learned jointly with the policy head parameters $\theta$. We train $(\theta, \phi)$ jointly by minimizing the negative log-likelihood (NLL) loss [1]:

$$\mathcal{L}_{\text{act}}(\theta, \phi) := \mathbb{E}\left[-\log \pi_\theta(A \mid S, \phi(S, G))\right]. \tag{7}$$

Importantly, the low-level policy accesses the full goal $G$ **only during training** to learn $(\theta, \phi)$. At deployment, we detach the encoder $\phi$, and the policy $\pi_\theta$ conditions directly on a subgoal representation $Z_{\text{sub}}$ predicted by the high-level policy, rather than on the true final goal.

As discussed in Section 4, we refer to the encoder learned via this policy optimization objective as an *actor-based goal representation* $\phi_A$. In contrast, standard Hierarchical methods for GCRL (Park et al., 2023; 2025d) typically learn *value-based representations* $\phi_V$ via value estimation objectives (*e.g.*, temporal difference learning).

## 8.3. Near-Optimal NLL Implies Approximate Action Sufficiency

We now show that learning a low-level policy formulated in this way through minimizing actor NLL naturally induces action-sufficient $\phi$. Detailed proofs are provided in Appendix G.

In Section 5, we defined the conditional KL risk in Eq. (3) as the fundamental quantity to minimize for obtaining the optimum low-level policy. The following lemma connects this theoretical risk to our tractable training objective.

---

**Lemma 8.1** (Decomposition of Actor NLL). *For any* $(\theta, \phi)$ *with* $Z = \phi(S, G)$*, the actor NLL* $\mathcal{L}_{\text{act}}(\theta, \phi)$ *decomposes as:*

$$\mathcal{L}_{\text{act}}(\theta, \phi) = H(A \mid S, G) + \mathcal{R}(\pi_\theta; \phi), \tag{8}$$

*in which* $\mathcal{R}(\pi_\theta; \phi)$ *is the conditional KL risk defined in Eq. (3).*

---

Crucially, the entropy term $H(A|S,G)$ is determined by the joint law $P$ established in Section 5 and is irreducible with respect to optimization. Therefore, minimizing the actor NLL $\mathcal{L}_{\text{act}}(\theta, \phi)$ is equivalent to minimizing the conditional KL risk $\mathcal{R}(\pi_\theta; \phi)$. By combining Lemma 8.1 with

---

[1] For ease of exposition, we assume expert data sampled from the optimal goal-conditioned action law $P(A \mid S, G)$. While direct access to such an oracle is unavailable, this is approximated in practice by AWR (2) objectives that concentrate probability mass on near-optimal actions.

---

the risk decomposition in Proposition 5.1, we obtain our main theoretical result: *achieving near-optimal Actor NLL guarantees that the learned representation is approximately action-sufficient.*

---

**Theorem 8.2** (Near-Optimal Actor NLL Implies Approximate Action Sufficiency). *For any* $(\theta, \phi)$ *with* $Z = \phi(S, G)$*, the action sufficiency gap* $\Delta_A = I(A; G \mid S, Z)$ *is bounded by the excess risk:*

$$\Delta_A \leq \mathcal{L}_{\text{act}}(\theta, \phi) - H(A \mid S, G). \tag{9}$$

*In particular, if the actor NLL is near-optimal such that* $\mathcal{L}_{\text{act}}(\theta, \phi) \leq H(A \mid S, G) + \varepsilon$*, then*

$$\Delta_A \leq \varepsilon, \tag{10}$$

*i.e.,* $Z$ *is* $\varepsilon$*-approximately action-sufficient.*

---

## 8.4. Theoretical Interpretation and Practical Implications

Theorem 8.2 implies that standard NLL minimization serves a dual purpose: it optimizes the low-level policy while simultaneously forcing the representation $\phi$ to retain all goal information necessary for action prediction. In particular, assuming the parameterized policy class in Eq. (6) has sufficient expressivity to approximate the optimal kernel $P(A \mid S, G)$, minimizing actor NLL drives the excess risk to zero, thereby ensuring $\Delta_A \to 0$.

While grounded in rigorous analysis, the proposed method is very simple in practice: we attach a goal encoder $\phi(S, G)$ directly to the input of the low-level policy and jointly minimize the actor NLL.

Our analysis establishes actor-based representations as a principled alternative for accurate low-level action prediction. By retaining the structural information necessary for action prediction, this approach avoids the potential information loss of value-based representations.

## 9. Experiment

We investigate the impact of various representations by integrating them into HIQL (Park et al., 2023), a versatile hierarchical framework for GCRL, using **OGBench** (Park et al., 2025a) as our testbed—a challenging benchmark for offline GCRL spanning both state-based and pixel-based observations. We compare against standard offline GCRL baselines GCIQL and GCIVL (Kostrikov et al., 2022), and report success rates averaged over multiple random seeds.

**Main results.** Tables 1 and 2 show that substituting the value-based representation ($\phi_V$) with our actor-based representation ($\phi_A$) yields consistent and substantial performance

*Table 1.* Success rate comparison of offline GCRL algorithms across state-based OGBench tasks. Results are averaged across 8 seeds.

| Method | Goal Rep. | cube-double | | cube-triple | | cube-quadruple | | scene |
|---|---|---|---|---|---|---|---|---|
| | | play | noisy | play | noisy | play | noisy | play |
| HIQL[OTA] | Value Rep. ($\phi_V$) | 0.26 | 0.25 | 0.16 | 0.09 | 0.01 | 0.02 | 0.56 |
| HIQL[OTA] | Actor Rep. ($\phi_A$) | 0.42 | 0.29 | 0.41 | 0.19 | **0.28** | 0.08 | 0.78 |
| H–Flow[OTA] | Value Rep. ($\phi_V$) | 0.26 | 0.28 | 0.15 | 0.10 | 0.01 | 0.01 | 0.69 |
| H–Flow[OTA] | Actor Rep. ($\phi_A$) | 0.43 | **0.33** | **0.45** | **0.23** | 0.25 | **0.11** | **0.81** |
| GCIQL | – | **0.69** | 0.13 | 0.30 | 0.14 | 0.01 | 0.00 | 0.53 |
| GCIVL | – | 0.36 | 0.20 | 0.08 | 0.14 | 0.00 | 0.00 | 0.54 |

*Table 2.* Success rate comparison on pixel-based OGBench tasks. Results are averaged over 3 seeds.

| Method | Goal Rep. | visual-cube | visual-scene |
|---|---|---|---|
| HIQL[OTA] | Value Rep. ($\phi_V$) | 0.05 | 0.23 |
| HIQL[OTA] | Actor Rep. ($\phi_A$) | **0.53** | **0.53** |
| GCIQL | – | 0.01 | 0.10 |
| GCIVL | – | 0.13 | 0.36 |

*Table 3.* Comparison of success rates across different goal representation choices on cube tasks.

| Goal Rep. | cube-double | | cube-triple | |
|---|---|---|---|---|
| | play | noisy | play | noisy |
| No Rep. | 0.27 | 0.24 | 0.23 | 0.06 |
| Autoencoder Rep. | 0.35 | – | 0.24 | – |
| Value Rep. ($\phi_V$) | 0.26 | 0.25 | 0.16 | 0.10 |
| Actor Rep. ($\phi_A$) | **0.42** | **0.29** | **0.41** | **0.19** |

gains across all environments. This gap is especially stark in tasks where value-based representations fail entirely (*e.g.*, cube-quadruple). Furthermore, to address the limited expressivity of unimodal Gaussian policies used in standard HIQL, we additionally employ flow matching (Kang et al., 2023; Park et al., 2025c) as a generative backbone for policies (Hierarchical Flow, H–Flow[OTA]), which further improves results by handling the multimodal action distributions required for these tasks. The advantage of actor-based representations becomes most pronounced under pixel-based observations (Table 2), where the abundance of nuisance information makes it more challenging to preserve action-relevant features through value learning alone.

**Necessity of goal representation.** The most straightforward approach is to condition the policy directly on the raw subgoal $s_{t+k}$ without a learned encoder (No Rep. in Table 3). While this theoretically guarantees maximum action sufficiency by preserving all information, Table 3 shows it is empirically suboptimal. As noted by Park et al. (2023), high-dimensional raw goals contain nuisance information that severely hinders high-level subgoal prediction.

An autoencoder-based representation, which jointly trains an encoder to compress $(s, g)$ into a low-dimensional latent $z$ and a decoder to reconstruct $g$ from $(s, z)$, only partially mitigates this issue: by forcing exact goal reconstruction, the latent $z$ still retains nuisance information that hinders high-level prediction. While value-based representations compress the noise, they ultimately degrade performance by discarding action-relevant features. In contrast, our actor-based representation resolves this trade-off by filtering out nuisance while strictly preserving the structure necessary for accurate low-level action.

## 10. Concluding Remarks

We formally defined **action sufficiency,** identifying it as a critical requirement for goal representations for hierarchical approaches in offline GCRL. We demonstrated that widely used value-based representations often discard essential action information, whereas our proposed **actor-based goal representations** are provably approximately action-sufficient. Empirical results on both the discrete domain and complex manipulation tasks confirm that satisfying this property significantly enhances hierarchical control performance.

Despite these positive results, several limitations point to important directions for future work.

**Gap between theory and practice.** Our theoretical analysis assumes access to the true distribution, whereas offline learning relies on static datasets. Consequently, limited data coverage may widen the gap between theoretical guarantees and practical action sufficiency.

**Sufficiency vs. predictability.** While action sufficiency is necessary for the low-level policy, it is not sufficient for the entire hierarchy. As the goal representation acts as an interface between planning and control, it must also be compact and predictable for the high-level policy, as shown in Table 3. Future work should explore balancing action-relevant information with the compressibility required for efficient high-level planning.

## Impact Statement

This paper presents work whose goal is to advance the field of machine learning, specifically goal representation learning in hierarchical offline reinforcement learning with primary applications in robotic manipulation. Our contribution is methodological and does not introduce capabilities beyond those addressed by existing hierarchical reinforcement learning methods. There are many potential societal consequences of our work, none of which we feel must be specifically highlighted here.

## Acknowledgments

This work was supported in part by the National Research Foundation of Korea (NRF) grant [No. RS-2025-02263628], the Institute of Information & communications Technology Planning & Evaluation (IITP) grants [RS-2021-II212068, RS-2022-II220113, RS-2022-II220959, RS-2021-II211343], and the BK21 FOUR Education and Research Program for Future ICT Pioneers (Seoul National University), funded by the Korean government. It was also supported by AOARD Grant No. FA2386-23-1-4079.

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

# Appendix

## A. Hierarchical Implicit Q-Learning (HIQL) & Temporal Abstraction.

A central challenge in GCRL is accurately estimating value functions for distant goals, which is crucial for solving long-horizon and temporally extended tasks (Huang et al., 2019; Kim et al., 2021; Park et al., 2023). To mitigate this difficulty, HIQL (Park et al., 2023) introduces a hierarchical policy framework built upon a value function learned via Implicit Q-Learning (IQL) (Kostrikov et al., 2022). This hierarchical structure allows the agent to select meaningful actions even when value estimates for faraway goals are inaccurate or noisy. However, as deeply studied in (Park et al., 2025b; Ahn et al., 2025), the value learning scheme in IQL failed to capture the long-horizon reward signal effectively. To solve this problem, (Park et al., 2025b; Ahn et al., 2025) adopt temporal abstraction scheme which updates the value over $n$ step target to decrease the effective horizon from start state $S$ to goal $g$.

Specifically, HIQL with temporal abstraction learns a goal-conditioned state-value function $V$ by minimizing the following expectile regression objective:

$$\mathcal{L}(V) = \mathbb{E}_{(s,s_n)\sim\mathcal{D},\, g\sim p(g)} \left[ L_2^\tau \left( r(s_n, g) + \gamma \bar{V}(s_n, g) - V(s, g) \right) \right], \tag{11}$$

where the expectile loss is defined as $L_2^\tau(u) = |\tau - \mathbf{1}(u < 0)|u^2$ with $\tau > 0.5$, and $\bar{V}$ denotes a target value network.[2]

Unlike prior studies (Andrychowicz et al., 2017; Chane-Sane et al., 2021; Wang et al., 2023; Park et al., 2023; Wu et al., 2024), we instead adopt an option-aware sparse goal-reaching reward (Ahn et al., 2025; Park et al., 2025b) of the form $r(s_n, g) = -\mathbf{1}\{s_n \neq g\}$, where $s_n$ denotes $n$ step forward state from the state $s$. Under this reward structure, the magnitude of the optimal value function $|V^\star(s, g)|$ corresponds to the *discounted temporal distance with temporal abstraction*, representing discounted estimate of the minimum number of environment steps required to reach goal $g$ from state $s$ with temporal abstraction.

HIQL decouples policy extraction into two hierarchical components. The high-level policy $\pi^h(s_{t+k} \mid s_t, g)$ proposes a $k$-step subgoal that guides progress toward the target goal, while the low-level policy $\pi^\ell(a_t \mid s_t, s_{t+k})$ generates primitive actions to reach the proposed subgoal. Both policies are learned using advantage-weighted regression (AWR) (Wang et al., 2018; Peng et al., 2019; Nair et al., 2020), with objectives given by

$$\mathcal{J}(\pi^h) = \mathbb{E}_{(s_t, s_{t+k}, g)\sim\mathcal{D}} \left[ \alpha^h \cdot \log \pi^h(s_{t+k} \mid s_t, g) \right], \tag{12}$$

$$\mathcal{J}(\pi^\ell) = \mathbb{E}_{(s_t, a_t, s_{t+1}, s_{t+k})\sim\mathcal{D}} \left[ \alpha^\ell \cdot \log \pi^\ell(a_t \mid s_t, s_{t+k}) \right], \tag{13}$$

where $\alpha^h = \exp\left(\beta^h A^h(s_t, s_{t+k}, g)\right)$, $\alpha^\ell = \exp\left(\beta^\ell A^\ell(s_t, s_{t+1}, s_{t+k})\right)$, $\beta^h$ and $\beta^\ell$ are inverse temperature parameters. The high-level advantage is defined as $A^h(s_t, s_{t+k}, g) = V^h(s_{t+k}, g) - V^h(s_t, g)$, and the low-level advantage as $A^\ell(s_t, s_{t+1}, s_{t+k}) = V^\ell(s_{t+1}, s_{t+k}) - V^\ell(s_t, s_{t+k})$. Note that, since the high level policy $\pi^h$ should plan the subgoal properly when the distance between $s$ and $g$ is quite large, and the low level policy $\pi^\ell$ only works on short-horizon scenario, we adopt large $n$ for $V^h$ and $n = 1$ for $V^\ell$.

## B. Additional Discussions

### B.1. Justification for State-Dependent Representation

In Section 5, we defined the goal representation as a function of both the current state and the goal, denoted by $Z = \phi(S, G)$. While goal-conditioned reinforcement learning often utilizes goal-only encoders (*i.e.*, $Z = \psi(G)$; see, *e.g.*, Park et al., 2025d), we argue that the state-dependent formulation is a more general and architecturally natural choice for hierarchical control.

We provide a detailed justification for this design choice based on three perspectives: mathematical subsumption, theoretical invariance, and architectural consistency.

#### B.1.1. MATHEMATICAL SUBSUMPTION

First and foremost, the set of state-dependent encoders strictly subsumes the set of goal-only encoders. A representation that depends solely on the goal is simply a special case of our formulation where the dependence on $S$ is trivial.

---

[2]Due to the overestimation issues inherent in IQL, we assume deterministic environment dynamics in this work.

Formally, let $\Psi$ be the set of measurable functions mapping $\mathcal{G} \to \mathcal{Z}$. For any goal-only encoder $\psi \in \Psi$, we can construct an equivalent state-dependent encoder $\phi$ as follows:

$$\phi(S, G) := (\psi \circ \Pi_{\mathcal{G}})(S, G),$$

where $\Pi_{\mathcal{G}} : \mathcal{S} \times \mathcal{G} \to \mathcal{G}$ is the standard canonical projection defined by $\Pi_{\mathcal{G}}(s, g) = g$.

This implies that our analysis of $\phi(S, G)$ entails no loss of generality; any result derived for the state-dependent case naturally applies to goal-only encoders by restricting the function class. By allowing $\phi$ to access $S$, we simply expand the hypothesis space for the representation, potentially allowing for more compact encodings (*e.g.*, relative goal coordinates) that are impossible with $\psi(G)$ alone.

### B.1.2. INVARIANCE OF ACTION SUFFICIENCY

One might ask whether introducing $S$ into $\phi$ complicates the definition of action sufficiency. We emphasize that our core definition is agnostic to the internal structure of $\phi$.

Recall the definition of action sufficiency:

$$I(A; G \mid S, Z) = 0.$$

This condition requires that $Z$, *in conjunction with* $S$, allows for the optimal prediction of $A$. Even if $Z$ were derived solely from $G$ (*i.e.*, $Z = \psi(G)$), the conditioning on $S$ in the mutual information term remains present. Therefore, the theoretical validity of our decomposition and the resulting sufficiency condition holds regardless of whether the encoder explicitly takes $S$ as input.

### B.1.3. ARCHITECTURAL CONSISTENCY IN HIERARCHICAL METHODS

Finally, from a practical standpoint, the state-dependent formulation aligns naturally with the information structure of Hierarchical methods for GCRL. In this framework, all major components inherently possess access to the current state $S$:

- The **High-Level Policy** $\pi^h(G_{\text{sub}} \mid S, G)$ observes $S$ to decide which subgoal is reachable.
- The **Low-Level Policy** $\pi^\ell(A \mid S, Z)$ observes $S$ to execute immediate actions.
- The **Value Function** $V(S, G)$ (or $V(S, Z)$) observes $S$ to estimate expected returns.

Since the policy side already conditions on $S$, "contextualizing" the encoder side with $S$ creates a symmetric information structure. For instance, in navigation tasks, encoding a goal as a "relative vector from the current state" (which requires access to both $S$ and $G$) is often far more efficient and robust than encoding absolute coordinates. Our formulation $\phi(S, G)$ formally accommodates such relative representations, whereas a rigid $\psi(G)$ formulation would exclude them.

### B.2. Limitations of Goal-Only Value-Based Representations

Prior work, *e.g.*, Park et al. (2023; 2025d), studies goal representations of the form $Z = \psi(G)$ constructed from value information, arguing that such value-based representations can be sufficient for predicting goal-conditioned behavior. Here, we analyze the limitations of this approach, specifically showing that under reasonable assumptions, strict value sufficiency for goal-only representations forces the retention of *all* goal information, including nuisance factors.

### B.2.1. ASSUMPTIONS AND DEFINITIONS

We assume the value target $V(s, g)$ uniquely identifies the goal when the agent is at the goal state.

**Assumption B.1** (Zero if and only if identity). For all $s, g \in \mathcal{S} = \mathcal{G}$,

$$V(s, g) = 0 \quad \Longleftrightarrow \quad s = g.$$

Previous works effectively utilize a condition slightly stronger than our information-theoretic value sufficiency, which we formalize as *strict value sufficiency*.

**Definition B.2** (Strict value sufficiency). A goal representation $Z = \psi(G)$ is *strictly value-sufficient* if there exists a measurable function $\tilde{v}$ such that

$$V(s, g) = \tilde{v}\big(s, \psi(g)\big) \qquad \text{for all } (s, g) \in \mathcal{S} \times \mathcal{G}.$$

B.2.2. INJECTIVITY AND INFORMATION EQUIVALENCE

Under Assumption B.1, strict value sufficiency implies that $\psi$ must be injective. This result formalizes the intuition that if two distinct goals map to the same representation, their value functions must be identical everywhere, leading to a contradiction at the goal states.

B.2.3. INJECTIVITY AND INFORMATION EQUIVALENCE

We first establish a general measure-theoretic lemma stating that an injective representation allows for the perfect reconstruction of the original variable, thereby generating the same $\sigma$-algebra.

**Lemma B.3** (Injectivity implies information equivalence). *Let $Z = \psi(G)$ where $\psi : \mathcal{G} \to \mathcal{Z}$ is a measurable function between standard Borel spaces. If $\psi$ is injective, then*

$$\sigma(G) = \sigma(Z) \quad \text{(modulo null sets)},$$

*where $\sigma(X)$ denotes the $\sigma$-algebra generated by the random variable $X$.*

*Proof.* Since $Z$ is a function of $G$, the inclusion $\sigma(Z) \subseteq \sigma(G)$ holds trivially. Conversely, for standard Borel spaces, an injective measurable function maps measurable sets to measurable sets and possesses a measurable inverse on its range. Thus, $G$ can be recovered from $Z$ almost surely via the inverse map $\psi^{-1}$, implying $\sigma(G) \subseteq \sigma(Z)$. Therefore, the two $\sigma$-algebras coincide. □

Using this lemma, we now show that under strict value sufficiency, the representation must preserve all goal information. The proof proceeds by first showing that strict value sufficiency forces $\psi$ to be injective.

**Proposition B.4** (Strict value sufficiency implies $\sigma(G) = \sigma(Z)$). *Assume $Z = \psi(G)$ is strictly value-sufficient. Then*

$$\sigma(G) = \sigma(Z) \qquad \text{(modulo null sets)}.$$

*Proof.* First, we show that $\psi$ is injective on $\mathcal{G}$. Suppose for the sake of contradiction that there exist distinct goals $g_1 \neq g_2$ such that $\psi(g_1) = \psi(g_2)$. By strict value sufficiency (Definition B.2), the value function must satisfy:

$$V(s, g_1) = \tilde{v}(s, \psi(g_1)) = \tilde{v}(s, \psi(g_2)) = V(s, g_2)$$

for all states $s \in \mathcal{S}$. Evaluating this at $s = g_1$, we have $V(g_1, g_1) = V(g_1, g_2)$. By Assumption B.1, $V(g_1, g_1) = 0$, which implies $V(g_1, g_2) = 0$. Applying Assumption B.1 again to the right-hand side necessitates $g_1 = g_2$, which contradicts our assumption that $g_1 \neq g_2$.

Thus, $\psi$ must be injective. Applying Lemma B.3, we conclude that $\sigma(G) = \sigma(Z)$. □

B.2.4. DISCUSSION

Proposition B.4 demonstrates that under strict value sufficiency, a goal-only representation $Z = \psi(G)$ must retain *exactly* the same information as the original goal $G$. This is often undesirable because high-dimensional goals typically contain nuisance information that is irrelevant for control. Forcing the representation to preserve all goal information contradicts the principle of abstraction in hierarchical learning, where the goal representation should ideally act as a compressed interface passing only relevant information to the low-level controller.

Unlike this rigid requirement, our proposed framework isolates *action-relevant* information via action sufficiency, allowing for many-to-one mappings that discard nuisance factors while preserving the control-relevant content necessary for optimal policy execution.

## C. Proofs for Section 5

In this section, we provide the formal derivations for the conditional KL risk decomposition (Proposition 5.1) and discuss the theoretical implications of action sufficiency.

### C.1. Goal-Marginalization Lemma

Before proving the main decomposition, we establish a useful lemma characterizing the relationship between the optimal policy conditioned on the raw goal $G$ and the optimal policy conditioned on the representation $Z$. This justifies viewing $P(\cdot \mid S, Z)$ as the posterior mixture of the expert policy.

**Lemma C.1** (Goal-Marginalization). *Let $Z = \phi(S, G)$ be a deterministic representation. The conditional distribution of the optimal action $A$ given $(S, Z)$ satisfies the following marginalization property:*

$$P(A \in B \mid S, Z) = \mathbb{E}\left[P(A \in B \mid S, G) \mid S, Z\right] \tag{14}$$

*for any measurable set $B \subseteq \mathcal{A}$.*

*Proof.* By the tower property of conditional expectation and noting that $Z$ is $\sigma(S, G)$-measurable (since $Z = \phi(S, G)$), we have:

$$\mathbb{E}[\mathbf{1}\{A \in B\} \mid S, Z] = \mathbb{E}\left[\mathbb{E}[\mathbf{1}\{A \in B\} \mid S, G, Z] \,\middle|\, S, Z\right]$$
$$= \mathbb{E}\left[\mathbb{E}[\mathbf{1}\{A \in B\} \mid S, G] \,\middle|\, S, Z\right].$$

The second equality holds because conditioning on $(S, G)$ fully determines $Z$, making $Z$ redundant in the inner expectation (i.e., $P(\cdot \mid S, G, Z) = P(\cdot \mid S, G)$). The left-hand side is $P(A \in B \mid S, Z)$ and the inner term on the right-hand side is $P(A \in B \mid S, G)$, which proves Eq. (14). $\qquad\square$

### C.2. Proof of the Conditional KL Risk Decomposition

We now provide the proof for the decomposition of the conditional KL risk. We first recall the statement of the proposition:

**Proposition C.2** (Conditional KL Risk Decomposition). *The risk $\mathcal{R}(\pi^\ell; \phi)$ decomposes into a modeling error term and a representation error term:*

$$\mathcal{R}(\pi^\ell; \phi) = \underbrace{\mathbb{E}\left[D_{KL}(P(A|S, Z) \,\|\, \pi^\ell(A|S, Z))\right]}_{\text{Modeling Error}}$$
$$+ \underbrace{I(A; G \mid S, Z)}_{\text{Representation Error}},$$

*where $I(\cdot; \cdot|\cdot)$ denotes conditional mutual information.*

*Proof.* Recall the definition of the risk from Eq. (3):

$$\mathcal{R}(\pi^\ell; \phi) = \mathbb{E}\left[D_{\text{KL}}(P(A|S, G) \,\|\, \pi^\ell(A|S, Z))\right].$$

We define the conditional mutual information term using its standard characterization. Since $Z = \phi(S, G)$ is deterministic, $P(A \mid S, G, Z) = P(A \mid S, G)$ almost surely, simplifying the mutual information to:

$$I(A; G \mid S, Z) = \mathbb{E}\left[D_{\text{KL}}(P(A \mid S, G) \,\|\, P(A \mid S, Z))\right]. \tag{15}$$

Now, consider the pointwise KL divergence inside the expectation of the risk $\mathcal{R}$. For fixed $(s, g, z)$, we decompose the log-likelihood ratio:

$$D_{\text{KL}}(P(\cdot|s, g) \,\|\, \pi^\ell(\cdot|s, z)) = \int P(a|s, g) \log \frac{P(a|s, g)}{\pi^\ell(a|s, z)} \, da$$
$$= \int P(a|s, g) \log \left(\frac{P(a|s, g)}{P(a|s, z)} \cdot \frac{P(a|s, z)}{\pi^\ell(a|s, z)}\right) da$$
$$= \underbrace{\int P(a|s, g) \log \frac{P(a|s, g)}{P(a|s, z)} \, da}_{\text{(I)}} + \underbrace{\int P(a|s, g) \log \frac{P(a|s, z)}{\pi^\ell(a|s, z)} \, da}_{\text{(II)}}.$$

We now take the expectation over $(S, G)$ (and implicitly $Z$):

1. **Term (I):** The expectation of the first term corresponds exactly to Eq. (15), which is the **Representation Error** $I(A; G \mid S, Z)$.

2. **Term (II):** For the second term, we apply the law of iterated expectations, conditioning on $(S, Z)$:

$$\mathbb{E}_{S,G} \left[ \int P(a|S,G) \log \frac{P(a|S,Z)}{\pi^\ell(a|S,Z)} \, da \right]$$

$$= \mathbb{E}_{S,Z} \left[ \mathbb{E}_{G|S,Z} \left[ \int P(a|S,G) \log \frac{P(a|S,Z)}{\pi^\ell(a|S,Z)} \, da \right] \right]$$

$$= \mathbb{E}_{S,Z} \left[ \int \underbrace{\mathbb{E}_{G|S,Z}[P(a|S,G)]}_{P(a|S,Z) \text{ by Lemma C.1}} \log \frac{P(a|S,Z)}{\pi^\ell(a|S,Z)} \, da \right]$$

$$= \mathbb{E}_{S,Z} \left[ D_{\mathrm{KL}}(P(\cdot|S,Z) \,\|\, \pi^\ell(\cdot|S,Z)) \right].$$

This yields the **Modeling Error**.

Combining these two terms completes the proof. $\qquad\square$

### C.3. Optimality of Action Sufficiency

The decomposition derived above allows us to formally state the conditions under which the control risk can be minimized to zero.

**Corollary C.3** (Zero Minimal KL Risk $\iff$ Action Sufficiency). *For a deterministic representation $Z = \phi(S, G)$, the following statements are equivalent:*

1. *The representation is action-sufficient: $I(A; G \mid S, Z) = 0$.*

2. *The minimum achievable risk is zero: $\inf_{\pi^\ell} \mathcal{R}(\pi^\ell; \phi) = 0$.*

*Proof.* From Proposition C.2, the risk is the sum of a non-negative modeling error and the representation error $\Delta_A = I(A; G \mid S, Z)$.

$$\inf_{\pi^\ell} \mathcal{R}(\pi^\ell; \phi) = \inf_{\pi^\ell} \left( \mathbb{E}\left[ D_{\mathrm{KL}}(P(A|S,Z) \,\|\, \pi^\ell(A|S,Z)) \right] \right) + \Delta_A.$$

The modeling error is minimized to 0 if and only if $\pi^\ell(\cdot \mid S, Z) = P(\cdot \mid S, Z)$ almost surely. Thus, the infimum of the risk is exactly $\Delta_A$. Consequently, the risk can be driven to zero if and only if $\Delta_A = 0$. $\qquad\square$

# D. Proofs for Section 6

In this section, we provide the formal proofs characterizing the limitations of value-based representations, specifically showing how value sufficiency implies functional dependence yet fails to guarantee action sufficiency.

### D.1. Equivalence of Value Sufficiency and Functional Dependence

We first establish that value sufficiency is equivalent to the condition that the optimal value $V$ can be perfectly reconstructed from the representation $(S, Z)$.

**Lemma D.1** (Characterization of Value Sufficiency). *Assume $V = v(S, G)$ and $Z = \phi_V(S, G)$ are deterministic measurable functions of $(S, G)$. Then the following are equivalent:*

(i) *$Z$ is value-sufficient, i.e. $I(V; G \mid S, Z) = 0$.*

(ii) *$V \perp G \mid (S, Z)$.*

(iii) *$V$ is $\sigma(S, Z)$-measurable, i.e. there exists a measurable map $\tilde{v}$ such that*

$$V = \tilde{v}(S, Z) \quad a.s. \tag{16}$$

*Proof.* **(i)** $\iff$ **(ii).** This follows immediately from the standard property of conditional mutual information: $I(X;Y \mid Z) = 0$ if and only if $X$ and $Y$ are conditionally independent given $Z$. Thus, $I(V;G \mid S,Z) = 0 \iff V \perp G \mid (S,Z)$.

**(ii)** $\implies$ **(iii).** Let $f : \mathbb{R} \to \mathbb{R}$ be any bounded measurable function. By the conditional independence assumption $V \perp G \mid (S,Z)$, we have:

$$\mathbb{E}\big[f(V) \mid S,Z,G\big] = \mathbb{E}\big[f(V) \mid S,Z\big] \quad \text{a.s.} \tag{17}$$

However, since $V = v(S,G)$ is a deterministic function of $(S,G)$, $V$ is measurable with respect to $\sigma(S,G)$. Consequently, conditioning on $(S,G)$ (and thus $(S,Z,G)$ since $Z$ is determined by $S,G$) reveals $V$ exactly:

$$\mathbb{E}\big[f(V) \mid S,Z,G\big] = f(V) \quad \text{a.s.} \tag{18}$$

Combining (17) and (18), we obtain:

$$f(V) = \mathbb{E}\big[f(V) \mid S,Z\big] \quad \text{a.s.}$$

Since the right-hand side is $\sigma(S,Z)$-measurable, $f(V)$ is $\sigma(S,Z)$-measurable for all bounded measurable $f$. Choosing $f$ as the identity (or appropriate indicators) implies $V$ is $\sigma(S,Z)$-measurable. By the Doob–Dynkin lemma, there exists a measurable function $\tilde{v}$ such that $V = \tilde{v}(S,Z)$ almost surely.

**(iii)** $\implies$ **(ii).** If $V = \tilde{v}(S,Z)$ almost surely, then $V$ is fully determined by $(S,Z)$. Therefore, for any measurable set $B$, $P(V \in B \mid S,Z,G) = \mathbf{1}_{\tilde{v}(S,Z) \in B} = P(V \in B \mid S,Z)$, which implies $V \perp G \mid (S,Z)$. $\square$

### D.2. General Decomposition of Action-Relevant Information

Here, we derive the general information-theoretic decomposition that relates action-relevant goal information to any intermediate representation $Z$ and the value signal $V$.

**Proposition D.2** (Exact Decomposition of Action Information). *Let $V = v(S,G)$ and $Z = \phi_V(S,G)$ be deterministic measurable functions of $(S,G)$. Then the following identity holds:*

$$I(A;G \mid S,Z) = I(A;G \mid S,V) \ - \ I(A;Z \mid S,V) \ + \ I(A;V \mid S,Z). \tag{19}$$

*Proof.* We derive the decomposition using the chain rule for conditional mutual information.

First, consider the term $I(A;G,V \mid S,Z)$. Since $V$ is a deterministic function of $(S,G)$, knowing $G$ (given $S$) determines $V$. Thus, adding $V$ to the target variables does not change the information:

$$I(A;G \mid S,Z) = I(A;G,V \mid S,Z).$$

Applying the chain rule to the pair $(G,V)$:

$$I(A;G,V \mid S,Z) = I(A;V \mid S,Z) + I(A;G \mid S,Z,V). \tag{20}$$

Next, we analyze the term $I(A;G \mid S,Z,V)$. Consider the mutual information $I(A;Z,G \mid S,V)$. We can expand this in two different ways using the chain rule:

**Expansion 1:** Condition on $Z$ first.

$$I(A;Z,G \mid S,V) = I(A;Z \mid S,V) + I(A;G \mid S,V,Z). \tag{21}$$

**Expansion 2:** Condition on $G$ first.

$$I(A;Z,G \mid S,V) = I(A;G \mid S,V) + I(A;Z \mid S,V,G).$$

Since $Z = \phi_V(S,G)$ is a deterministic function of $(S,G)$, given $(S,V,G)$, $Z$ is constant. Thus, $I(A;Z \mid S,V,G) = 0$, yielding:

$$I(A;Z,G \mid S,V) = I(A;G \mid S,V). \tag{22}$$

Equating (21) and (22), we have:

$$I(A;Z \mid S,V) + I(A;G \mid S,V,Z) = I(A;G \mid S,V).$$

Rearranging for the term appearing in (20):

$$I(A; G \mid S, Z, V) = I(A; G \mid S, V) - I(A; Z \mid S, V).$$

Substituting this back into (20), we obtain the final result:

$$I(A; G \mid S, Z) = I(A; V \mid S, Z) + I(A; G \mid S, V) - I(A; Z \mid S, V).$$

$\square$

## D.3. The Information Gap under Value Sufficiency

We now show that if a representation is value-sufficient, the decomposition simplifies, explicitly highlighting the irreducible information gap caused by value level sets.

**Corollary D.3** (Decomposition under Value Sufficiency). *If $Z$ is value-sufficient, then $I(A; V \mid S, Z) = 0$ and hence*

$$I(A; G \mid S, Z) = I(A; G \mid S, V) \; - \; I(A; Z \mid S, V). \tag{23}$$

*Proof.* By Lemma D.1, if $Z$ is value-sufficient, there exists a measurable function $\tilde{v}$ such that $V = \tilde{v}(S, Z)$ almost surely. This implies that $V$ is fully determined by $(S, Z)$. Consequently, $V$ contains no additional information about $A$ given $(S, Z)$, so $I(A; V \mid S, Z) = 0$. Substituting this into the identity from Proposition D.2 yields the stated result. $\square$

## E. A Sufficient Condition for $I(A; G \mid S, V) > 0$

In Section 6, we argued that conditioning on the optimal value $V$ is typically insufficient to resolve goal-dependent action ambiguity, implying $I(A; G \mid S, V) > 0$. In this appendix, we formalize this claim by providing a purely stochastic sufficient condition based on *conditional action variance*.

### E.1. Formal Setup and Assumption

Let $S, G, A, V$ be random variables on a probability space $(\Omega, \mathcal{F}, \mathbb{P})$ taking values in measurable spaces $(\mathcal{S}, \mathcal{B}(\mathcal{S}))$, $(\mathcal{G}, \mathcal{B}(\mathcal{G}))$, $(\mathcal{A}, \mathcal{B}(\mathcal{A}))$, and $(\mathbb{R}, \mathcal{B}(\mathbb{R}))$. In this section, we use $\mathbb{P}$ to explicitly denote the underlying probability measure, whereas the main text overloads $P$ for (conditional) laws and distribution functions.

**Assumption E.1** (Positive Conditional Action Variance). There exists a family of measurable action events

$$\{B_{s,v} \subseteq \mathcal{A} : (s, v) \in \mathcal{S} \times \mathbb{R}\}$$

such that the induced random event $\{A \in B_{S,V}\}$ satisfies:

$$\mathbb{E}\Big[\mathrm{Var}\big(\mathbb{P}(A \in B_{S,V} \mid S, V, G) \mid S, V\big)\Big] > 0. \tag{24}$$

Here, each $B_{s,v} \in \mathcal{B}(\mathcal{A})$ represents an action set dependent on the state and value.

### E.2. Intuition and Plausibility

**Intuitive Meaning.** This assumption captures the fundamental limitation of value as a scalar signal: $V$ typically encodes "distance" or "difficulty," but discards "direction." Consider an agent at a fork in the road; two goals may be equidistant (same $V$) yet require opposite actions (different $A$). The assumption serves as a stochastic detector for this phenomenon, checking if the action distribution fluctuates across goals within an iso-value set. If it does, $V$ has failed to screen off $G$ from $A$.

**Plausibility in Non-Trivial Environments.** This condition is satisfied in almost any environment beyond simple 1D tasks.

- **Symmetry:** Spatial symmetries naturally create the "fork" scenario described above (e.g., targets $N$ steps away in opposite directions have identical values but distinct optimal actions).

- **General Case:** In high-dimensional state spaces, value level sets are typically large manifolds that almost invariably encompass goals requiring different optimal actions, unless the value function is pathologically unique for every goal-action pair.

**E.3. Lower Bound for $I(A; G \mid S, V)$**

We now show that positive conditional action variance implies a positive information gap.

**Proposition E.2** (Positive Conditional Action Variance $\Rightarrow I(A; G \mid S, V) > 0$). *Under Assumption E.1, the conditional mutual information is strictly positive and lower-bounded by the conditional action variance:*

$$I(A; G \mid S, V) \geq 2\mathbb{E}\Big[\mathrm{Var}\big(\mathbb{P}(A \in B_{S,V} \mid S, V, G) \mid S, V\big)\Big] > 0. \tag{25}$$

*Proof.* We utilize Pinsker's inequality for Bernoulli distributions. For $\mathrm{Bern}(p)$ and $\mathrm{Bern}(q)$, Pinsker's inequality implies:

$$\mathrm{KL}(\mathrm{Bern}(p) \,\|\, \mathrm{Bern}(q)) \geq 2(p - q)^2. \tag{26}$$

Define the binary random variable indicating whether the action falls into the target set:

$$Y := \mathbf{1}\{A \in B_{S,V}\} \in \{0, 1\}.$$

Since $Y$ is a deterministic function of $(S, V, A)$, the data processing inequality for conditional mutual information gives:

$$I(A; G \mid S, V) \geq I(Y; G \mid S, V). \tag{27}$$

Now, let $p$ be the goal-dependent probability and $\bar{p}$ be the marginalized probability (averaged over goals):

$$p := \mathbb{P}(Y = 1 \mid S, V, G) = \mathbb{P}(A \in B_{S,V} \mid S, V, G),$$

$$\bar{p} := \mathbb{P}(Y = 1 \mid S, V) = \mathbb{E}[p \mid S, V].$$

Conditioned on $(S, V, G)$, $Y \sim \mathrm{Bern}(p)$, and conditioned on $(S, V)$ alone, the marginal distribution is $Y \sim \mathrm{Bern}(\bar{p})$. Expressing $I(Y; G \mid S, V)$ as the expected KL divergence:

$$I(Y; G \mid S, V) = \mathbb{E}\Big[\mathrm{KL}\big(\mathrm{Bern}(p) \,\|\, \mathrm{Bern}(\bar{p})\big)\Big].$$

Applying Pinsker's inequality (26) pointwise:

$$I(Y; G \mid S, V) \geq \mathbb{E}\Big[2(p - \bar{p})^2\Big].$$

Finally, recognizing that $\mathbb{E}[(p - \bar{p})^2 \mid S, V]$ is precisely the conditional variance of $p$:

$$\mathbb{E}\big[(p - \bar{p})^2\big] = \mathbb{E}\Big[\mathrm{Var}(p \mid S, V)\Big] = \mathbb{E}\Big[\mathrm{Var}\big(\mathbb{P}(A \in B_{S,V} \mid S, V, G) \mid S, V\big)\Big].$$

Combining this with (27) yields the bound in (25). The strict positivity follows directly from Assumption E.1. $\square$

# F. Discrete Cube: Detailed Setup

This appendix provides comprehensive details on the Discrete Cube environment introduced in Section 7, including the full specification of states, actions, transitions, the computation of optimal values and policies, the construction of the representation library, and the evaluation protocol.

### F.1. Environment Specification

**State space.** The environment is a $4 \times 4$ grid containing an agent and two cubes (Red and Blue). A state is defined as:

$$s = (p_a, p_r, p_b, h) \in \mathcal{S},$$

where:

- $p_a = (x_a, y_a) \in \{0, 1, 2, 3\}^2$ is the agent position,

- $p_r = (x_r, y_r)$ is the red cube position (or a special "held" marker when held),

- $p_b = (x_b, y_b)$ is the blue cube position (or a special "held" marker when held),

- $h \in \{\texttt{none}, \texttt{red}, \texttt{blue}\}$ is the gripper status.

When $h = \texttt{red}$, the red cube moves with the agent so that $p_r = p_a$; similarly for $h = \texttt{blue}$. The two cubes cannot occupy the same position when both are on the floor, i.e., $p_r \neq p_b$ when $h = \texttt{none}$.

For $n = 4$, the state space contains 4,352 reachable states.

**Action space.** The action set consists of six discrete actions:

$$\mathcal{A} = \{\texttt{up}, \texttt{down}, \texttt{left}, \texttt{right}, \texttt{pick}, \texttt{place}\}.$$

**Transition dynamics.** Transitions are deterministic:

- **Movement actions** (`up`, `down`, `left`, `right`): Move the agent by one cell in the specified direction, clipped at grid borders. If the agent is holding a cube, the cube moves with the agent.

- **`pick`**: Succeeds if and only if $h = \texttt{none}$ and the agent is on a cube. The agent picks up the cube at its current position.

- **`place`**: Succeeds if and only if $h \neq \texttt{none}$ and the agent's current position is not occupied by another cube. The held cube is placed at the agent's position.

**Goal space.** A goal specifies a target cube and a desired position:

$$g = (t, p_g) \in \mathcal{G}, \qquad t \in \{\texttt{red}, \texttt{blue}\}, \ p_g \in \{0, 1, 2, 3\}^2.$$

For $n = 4$, we have $|\mathcal{G}| = 2 \times 16 = 32$ goals.

**Success condition.** A state $s$ is successful for goal $g = (t, p_g)$ if and only if:

1. The target cube $t$ is at position $p_g$, and

2. The agent is not holding the target cube ($h \neq t$).

**State-goal pair filtering.** To ensure well-defined optimal actions and exclude trivial or degenerate cases, we filter the set of $(s, g)$ pairs used for computing information-theoretic quantities:

- When $h = \texttt{none}$: the red cube, blue cube, and goal positions must all be mutually distinct ($p_r \neq p_b$, $p_r \neq p_g$, $p_b \neq p_g$).

- When $h \neq \texttt{none}$: the agent position, floor cube position, and goal position must all be mutually distinct.

After filtering, we obtain 120,960 valid $(s, g)$ pairs for $n = 4$.

### F.2. Optimal Value and Policy via BFS

**Shortest-path distance.** For each goal $g$, we compute the exact optimal distance-to-success $D^\star(s, g) \in \mathbb{N}$ for all states $s$ using breadth-first search (BFS) backward from the success states:

1. Initialize $D^\star(s, g) = 0$ for all success states (where target cube is at $p_g$ and $h \neq t$).

2. Propagate distances backward through the transition graph: for each state $s'$ with an action $a$ leading to state $s$, set $D^\star(s', g) = D^\star(s, g) + 1$ if not yet visited.

3. Continue until all reachable states are labeled.

**Optimal value function.** We define the optimal value as the negative distance:

$$V^\star(s, g) := -D^\star(s, g).$$

This represents the (negated) minimum number of steps required to achieve the goal from state $s$.

For intuition, when not holding any cube, the distance roughly satisfies:

$$D^\star(s, g) \approx d_{at}(s, g) + d_{bg}(s, g) + 2,$$

where $d_{at}$ is the Manhattan distance from agent to target cube, $d_{bg}$ is the Manhattan distance from target cube to goal, and $+2$ accounts for `pick` and `place` actions. The exact BFS computation handles edge cases where this approximation breaks down (e.g., when holding the non-target cube).

**Optimal action distribution.** An action $a$ is optimal at state $s$ for goal $g$ if and only if executing $a$ leads to a successor state $s'$ with:

$$D^\star(s', g) = D^\star(s, g) - 1 \qquad \Leftrightarrow \qquad V^\star(s', g) = V^\star(s, g) + 1.$$

Let $\mathcal{A}^\star(s, g) \subseteq \mathcal{A}$ denote the set of optimal actions. We define the optimal action distribution with uniform tie-breaking:

$$P^\star(a \mid s, g) = \begin{cases} \frac{1}{|\mathcal{A}^\star(s,g)|} & \text{if } a \in \mathcal{A}^\star(s, g), \\ 0 & \text{otherwise.} \end{cases}$$

### F.3. Feature Representation

For constructing goal representations, we define the following features based on relative positions. Let $(x_a, y_a)$ be the agent position, $(x_t, y_t)$ the target cube position, and $(x_g, y_g)$ the goal position.

**Directional features.**

$$
\begin{aligned}
dx_1 &= x_t - x_a, & dy_1 &= y_t - y_a, \\
dx_2 &= x_g - x_t, & dy_2 &= y_g - y_t.
\end{aligned}
$$

Here $(dx_1, dy_1)$ points from the agent to the target cube, and $(dx_2, dy_2)$ points from the target cube to the goal.

**Distance features.**

$$
\begin{aligned}
d_{at} &= |dx_1| + |dy_1|, \\
d_{bg} &= |dx_2| + |dy_2|.
\end{aligned}
$$

**Handling held cubes.** When the agent is holding the target cube, we set $(x_t, y_t) = (x_a, y_a)$ so that $dx_1 = dy_1 = 0$ and $d_{at} = 0$.

### F.4. Representation Library

We construct a diverse library of goal representations $\phi : \mathcal{S} \times \mathcal{G} \to \mathcal{Z}$ to systematically explore the $(\Delta_V, \Delta_A)$ space.

F.4.1. BASELINE REPRESENTATIONS

We define four baseline representations that anchor different regions of the sufficiency space:

1. **full**: $Z = (h, t, dx_1, dy_1, dx_2, dy_2)$.

   Preserves complete directional information. This representation is both action-sufficient ($\Delta_A = 0$) and value-sufficient ($\Delta_V = 0$) by construction, as it retains all information needed to determine both the optimal action and the optimal value.

2. **signs**: $Z = (h, t, \text{sgn}(dx_1), \text{sgn}(dy_1), \text{sgn}(dx_2), \text{sgn}(dy_2))$.

   Coarsens directions to signs only $(-1, 0, +1)$, discarding magnitude information. This representation is nearly action-sufficient ($\Delta_A \approx 0.0003$) because directional signs are typically sufficient to determine the optimal movement direction. However, it is value-insufficient ($\Delta_V \approx 0.99$) because different magnitudes correspond to different distances (and hence values).

3. **value_only**: $Z = (V^\star, )$.

   Preserves only the optimal value. By construction, this is perfectly value-sufficient ($\Delta_V = 0$). However, it is highly action-insufficient ($\Delta_A \approx 0.23$) because many different state-goal pairs share the same value while requiring different optimal actions.

4. **distances**: $Z = (h, d_{at}, d_{bg})$.

   Preserves distances but not directions. This is value-sufficient ($\Delta_V = 0$) since the value depends only on distances. However, it is action-insufficient ($\Delta_A \approx 0.08$) because different directions with the same distances require different movement actions.

### F.4.2. RANDOM REPRESENTATION GENERATION

To broadly cover the $(\Delta_V, \Delta_A)$ plane, we generate 2000 random representations using a template-based approach. Each random representation is constructed by:

1. Sampling a template type from a predefined set with specified probabilities.

2. Sampling template-specific parameters (e.g., which features to include, which transforms to apply).

**Feature transforms.** We define several transforms that can be applied to individual features:

*Directional transforms* (applied to $dx_1, dy_1, dx_2, dy_2$):

- `raw`: identity, $x \mapsto x$

- `sign`: sign function, $x \mapsto \text{sgn}(x) \in \{-1, 0, +1\}$

- `abs`: absolute value, $x \mapsto |x|$

- `clip_k`: clip to $[-k, k]$, i.e., $x \mapsto \max(-k, \min(k, x))$ for $k \in \{1, 2, 3\}$

- `parity`: parity of absolute value, $x \mapsto |x| \mod 2$

- `sgn_bucket_k`: signed bucket, $x \mapsto \text{sgn}(x) \cdot \min(|x|, k)$ for $k \in \{2, 3\}$

*Distance transforms* (applied to $d_{at}, d_{bg}$):

- `raw`: identity

- `bucket_k`: $x \mapsto \min(x, k)$ for $k \in \{2, 3, 4\}$

- `parity`: $x \mapsto x \mod 2$

*Value transforms* (applied to $V^\star$):

- `raw`: identity

- `bucket_3`: bucket the cost $-V^\star$ into $\{0, 1, 2, 3, \geq 4\}$

**Template types.** The random representations are generated from the following templates:

- **value_plus**: Includes holding status, optionally target identity, and a transformed value. Tends to be value-sufficient with varying action sufficiency.

- **dist_coarse**: Includes holding status, optionally target identity, and transformed distances $d_{at}$, $d_{bg}$. Explores the value-sufficient but action-insufficient region.

- **dir_subset**: Includes holding status, optionally target identity, and a random subset of directional signs. Explores action-sufficient but value-insufficient region.

- **dir_coarse**: Includes all four directional features with independently sampled transforms. Covers a wide range of both sufficiency measures.

- **mixed_dir_dist**: Combines a subset of directional features with distance features, each with random transforms.

- **phase_split**: Conditions representation on the gripper status (holding vs. not holding), using different feature subsets for each phase.

- **proj_mod**: Projects directional features through random linear combinations modulo a small integer.

- **two_hash**: Hashes pairs of directional features independently.

- **hashed_actor**: Hashes the full directional tuple modulo a random integer, introducing collisions that degrade both sufficiency measures.

- **hashed_dist**: Hashes the distance tuple modulo a random integer.

- **drop_id**: Drops target identity, forcing the representation to be shared across red and blue cube goals.

Templates are sampled with weights designed to cover the $(\Delta_V, \Delta_A)$ plane broadly.

### F.5. Mixed Policy and Control Evaluation

**Mixed policy.** Given a representation $\phi$, the mixed policy conditions only on the current state $s$ and the representation value $z = \phi(s, g)$:

$$\pi_\phi(a \mid s, z) := \mathbb{E}[P^\star(a \mid s, G) \mid S = s, Z = z] = \sum_{g:\phi(s,g)=z} P^\star(a \mid s, g) \cdot P(G = g \mid S = s, Z = z).$$

Under the uniform distribution over filtered $(s, g)$ pairs, this simplifies to averaging $P^\star(\cdot \mid s, g)$ uniformly over all goals $g$ that map to the same $z$:

$$\pi_\phi(a \mid s, z) = \frac{1}{|\{g : \phi(s, g) = z\}|} \sum_{g:\phi(s,g)=z} P^\star(a \mid s, g).$$

**Rollout procedure.** To evaluate control success, we sample tasks and execute rollouts:

1. **Task sampling**: Sample 600 tasks $(s_0, g)$ where $h(s_0) = \texttt{none}$ and the target is not already at the goal.

2. **Horizon setting**: For each task, set $H = \min\{D^\star(s_0, g) + 6, 30\}$, where the margin $m = 6$ and cap $H_{\max} = 30$ prevent indefinite wandering while allowing some slack for suboptimal actions.

3. **Rollout**: For each task, execute 50 independent rollouts. At each step $t$:
   (a) Compute $z_t = \phi(s_t, g)$.
   (b) Sample $a_t \sim \pi_\phi(\cdot \mid s_t, z_t)$.
   (c) Execute $a_t$ to obtain $s_{t+1}$.
   (d) Terminate if success or $t \geq H$.

4. **Success rate**: For each representation $\phi$, compute the fraction of rollouts that reach success.

## F.6. Information-Theoretic Quantities

All information-theoretic quantities are computed exactly under the uniform distribution over filtered $(s, g)$ pairs.

**Conditional entropies.**

$$H(A \mid S, G) = \mathbb{E}_{s,g}\left[-\sum_a P^\star(a \mid s, g) \log P^\star(a \mid s, g)\right],$$

$$H(A \mid S, Z) = \mathbb{E}_{s,g}\left[-\sum_a \pi_\phi(a \mid s, \phi(s, g)) \log \pi_\phi(a \mid s, \phi(s, g))\right],$$

$$H(V \mid S, G) = 0,$$

$$H(V \mid S, Z) = \mathbb{E}_{s,g}\left[H(P(V \mid S = s, Z = \phi(s, g)))\right].$$

**Sufficiency gaps.**

$$\Delta_A := I(A; G \mid S, Z) = H(A \mid S, Z) - H(A \mid S, G),$$

$$\Delta_V := I(V; G \mid S, Z) = H(V \mid S, Z) - H(V \mid S, G) = H(V \mid S, Z).$$

**Within-level-set disambiguation.**

$$I(A; Z \mid S, V) = H(A \mid S, V) - H(A \mid S, V, Z),$$

where $H(A \mid S, V)$ is computed by grouping $(s, g)$ pairs by $(s, V^\star(s, g))$ and averaging the entropy of the optimal action distribution within each group.

# G. Proofs for Section 8

In this section, we provide the formal derivations connecting the actor training objective (log-loss) to the information-theoretic concept of action sufficiency.

## G.1. Decomposition of Actor Log-Loss

We first prove that the actor's negative log-likelihood (NLL) objective is equivalent to the conditional entropy of the optimal policy plus the conditional KL risk defined in the previous sections.

**Lemma G.1** (Actor Log-Loss as Excess KL Risk). *For any actor parameters $(\theta, \phi)$ with representation $Z = \phi(S, G)$, the actor training loss satisfies:*

$$\mathcal{L}_{\mathrm{act}}(\theta, \phi) = H(A \mid S, G) + \mathcal{R}(\pi_\theta, \phi), \tag{28}$$

*where $\mathcal{R}(\pi_\theta, \phi)$ is the conditional KL risk.*

*Proof.* Recall the definition of the conditional entropy of the data generating distribution:

$$H(A \mid S, G) = \mathbb{E}_{(S,G,A)\sim P}\left[-\log P(A \mid S, G)\right].$$

We expand the expression for the difference between the actor loss and the conditional entropy:

$$\mathcal{L}_{\mathrm{act}}(\theta, \phi) - H(A \mid S, G) = \mathbb{E}\left[-\log \pi_\theta(A \mid S, Z)\right] - \mathbb{E}\left[-\log P(A \mid S, G)\right]$$

$$= \mathbb{E}\left[\log P(A \mid S, G) - \log \pi_\theta(A \mid S, Z)\right]$$

$$= \mathbb{E}\left[\log \frac{P(A \mid S, G)}{\pi_\theta(A \mid S, Z)}\right].$$

By the law of iterated expectations, conditioning on $(S, G)$:

$$\mathbb{E}\left[\log \frac{P(A \mid S, G)}{\pi_\theta(A \mid S, Z)}\right] = \mathbb{E}_{(S,G)\sim P}\left[\mathbb{E}_{A\sim P(\cdot|S,G)}\left[\log \frac{P(A \mid S, G)}{\pi_\theta(A \mid S, Z)} \,\middle|\, S, G\right]\right]$$

$$= \mathbb{E}_{(S,G)\sim P}\Big[\mathrm{KL}\big(P(\cdot \mid S,G) \,\|\, \pi_\theta(\cdot \mid S,Z)\big)\Big]$$
$$= \mathcal{R}(\pi_\theta, \phi).$$

This completes the proof. □

### G.2. Bounding the Action Sufficiency Gap

Using the decomposition derived above, we now show that minimizing the actor log-loss implicitly minimizes the irreducible information gap $I(A; G \mid S, Z)$.

**Theorem G.2** (Near-Optimal Actor NLL Implies Approximate Action Sufficiency). *For any $(\theta, \phi)$ with $Z = \phi(S,G)$, the action sufficiency gap is bounded by the excess loss:*

$$I(A; G \mid S, Z) \;\leq\; \mathcal{L}_{\mathrm{act}}(\theta, \phi) \;-\; H(A \mid S, G). \tag{29}$$

*Consequently, if $\mathcal{L}_{\mathrm{act}}(\theta, \phi) \leq H(A \mid S, G) + \varepsilon$, then $Z$ is $\varepsilon$-approximately action-sufficient.*

*Proof.* From Lemma G.1, we have the identity:

$$\mathcal{L}_{\mathrm{act}}(\theta, \phi) - H(A \mid S, G) = \mathcal{R}(\pi_\theta, \phi). \tag{30}$$

Recall the decomposition of the conditional KL risk derived in the previous section (Proposition regarding Conditional KL Risk Decomposition):

$$\mathcal{R}(\pi_\theta, \phi) = \underbrace{\mathbb{E}\Big[\mathrm{KL}\big(P(A \mid S, Z) \,\|\, \pi_\theta(A \mid S, Z)\big)\Big]}_{\text{Modeling Error}\geq 0} + \underbrace{I(A; G \mid S, Z)}_{\text{Representation Error}}.$$

Since the KL divergence is non-negative, the modeling error term is non-negative. Therefore:

$$\mathcal{R}(\pi_\theta, \phi) \geq I(A; G \mid S, Z). \tag{31}$$

Combining Eq. (30) and Eq. (31) yields:

$$I(A; G \mid S, Z) \leq \mathcal{R}(\pi_\theta, \phi) = \mathcal{L}_{\mathrm{act}}(\theta, \phi) - H(A \mid S, G).$$

For the second part of the theorem, substituting the condition $\mathcal{L}_{\mathrm{act}}(\theta, \phi) \leq H(A \mid S, G) + \varepsilon$ into the inequality gives:

$$I(A; G \mid S, Z) \leq (H(A \mid S, G) + \varepsilon) - H(A \mid S, G) = \varepsilon.$$

Thus, $Z$ is $\varepsilon$-approximately action-sufficient. □

## H. Experimental Details

### H.1. Tasks and Datasets

We evaluate our approach using the `Cube` environment, which is a high-dimensional robotic manipulation suite. The task requires a robotic arm to manipulate and stack cubic blocks into specific goal configurations, presenting challenges in precision and long-horizon planning. To assess the scalability of our method, we conduct experiments across three levels of task complexity based on the number of objects: `double` (2 cubes), `triple` (3 cubes), and `quadruple` (4 cubes).

For each task variation, we utilize two distinct types of datasets representing different data quality distributions. The `play` dataset is generated via an oracle planner that demonstrates near-optimal behavior and efficient paths toward objectives. In contrast, the `noisy` dataset is collected using a markovian planner with added stochasticity, resulting in sub-optimal and exploratory trajectories. The data collection process is entirely task-agnostic; agents perform random interactions by picking a cube and placing it at a random coordinate. Each resulting dataset consists of 100M transitions, structured as 100 sub-datasets containing 1,000 trajectories of 1,000 timesteps each.

## H.2. Training and Evaluation Details

All offline GCRL algorithms are trained for a total of 2.5M gradient steps. We conduct intermediate evaluations every 500K steps to monitor the stability and progress of the learning process.

Success is measured against 5 specific goal configurations defined for each environment. During the evaluation phase, the agent is tested 20 times for each goal, resulting in 100 total trials per evaluation checkpoint. We report the average success rate across these trials. The specific hyperparameter configurations used for these experiments are detailed in Table 4.

*Table 4.* Common hyperparameters for experiments.

| Hyperparameter | Value |
|---|---|
| Learning rate | 3e-4 |
| Optimizer | Adam |
| Minibatch size | 1024 |
| Gradient steps | 2.5M |
| MLP size | [512, 512, 512] |
| Activation function | GELU |
| Target network update rate | 0.005 |
| Discount factor $\gamma$ | 0.995 |
| Horizon reduction factor $n$ | 1 (`cube-double`, `cube-triple`), 5 (`cube-quadruple`) |
| Subgoal steps $k$ | 25 |
| Expectile $\kappa$ | 0.7 (HIQL, OTA), 0.9 (GCIQL, GCIVL) |
| Policy extraction temperature $\alpha$ | 3.0 (HIQL, OTA), 1.0 (GCIQL), 10.0 (GCIVL) |
| Goal representation dimension | 10 |
| Actor $(p_{\text{cur}}^{\mathcal{D}}, p_{\text{traj}}^{\mathcal{D}}, p_{\text{rand}}^{\mathcal{D}})$ ratio | (0, 1, 0) |
| Value $(p_{\text{cur}}^{\mathcal{D}}, p_{\text{traj}}^{\mathcal{D}}, p_{\text{rand}}^{\mathcal{D}})$ ratio | (0.2, 0.5, 0.3) |

## H.3. Full Results

We report full results in Table 5, including mean and standard deviation of success rates across 8 seeds.

*Table 5.* Full success rate comparison across all methods and environments. Results are reported as mean $\pm$ std. **Bold** numbers indicate results at or above 95% of the best performance per row. Cube tasks and `scene-play` are averaged over 8 seeds; visual tasks are averaged over 3 seeds.

| Environment | HIQL$^{\text{OTA}}$ | | H–Flow$^{\text{OTA}}$ | | GCIQL | GCIVL |
|---|---|---|---|---|---|---|
| | Value Rep. $(\phi_V)$ | Actor Rep. $(\phi_A)$ | Value Rep. $(\phi_V)$ | Actor Rep. $(\phi_A)$ | | |
| `cube-double-play` | 0.26 ±0.10 | 0.42 ±0.03 | 0.26 ±0.04 | 0.43 ±0.05 | **0.69 ±0.03** | 0.36 ±0.05 |
| `cube-double-noisy` | 0.25 ±0.02 | 0.29 ±0.03 | 0.28 ±0.04 | **0.33 ±0.04** | 0.13 ±0.01 | 0.20 ±0.03 |
| `cube-triple-play` | 0.16 ±0.03 | 0.41 ±0.04 | 0.15 ±0.03 | **0.45 ±0.05** | 0.30 ±0.04 | 0.08 ±0.02 |
| `cube-triple-noisy` | 0.09 ±0.03 | 0.19 ±0.04 | 0.10 ±0.01 | **0.23 ±0.04** | 0.14 ±0.05 | 0.14 ±0.03 |
| `cube-quadruple-play` | 0.01 ±0.01 | **0.28 ±0.05** | 0.01 ±0.01 | 0.25 ±0.08 | 0.01 ±0.01 | 0.00 ±0.00 |
| `cube-quadruple-noisy` | 0.02 ±0.01 | 0.08 ±0.02 | 0.01 ±0.01 | **0.11 ±0.03** | 0.00 ±0.00 | 0.00 ±0.00 |
| `scene-play` | 0.56 ±0.07 | **0.78 ±0.04** | 0.69 ±0.05 | **0.81 ±0.05** | 0.53 ±0.05 | 0.54 ±0.06 |
| `visual-cube-play` | 0.05 ±0.03 | **0.53 ±0.06** | 0.07 ±0.04 | 0.01 ±0.01 | 0.01 ±0.01 | 0.13 ±0.01 |
| `visual-scene-play` | 0.23 ±0.07 | **0.53 ±0.02** | 0.19 ±0.15 | 0.07 ±0.01 | 0.10 ±0.01 | 0.36 ±0.06 |

# I. Q-Function-Based Representation

This appendix supplements the main text with a discussion of Q-function-based goal representations, which are also widely adopted in offline GCRL (Li et al., 2025; 2026; Song et al., 2026). We define Q-sufficiency, show that it implies action-sufficiency under a mild assumption, and empirically verify that a Q-based representation achieves performance comparable to our actor-based representation. Together, these results indicate that Q-based representations are themselves an instance of action-sufficient representations within our framework, supporting our broader thesis that the key property is action sufficiency rather than the specific learning objective.

## I.1. Q-Sufficiency Implies Action-Sufficiency

We first define the optimal goal-conditioned Q-function. With a discount factor $\gamma \in (0, 1]$ and reward function $r(s, g)$, the optimal Q-function is

$$Q^\star(s, a, g) = \sup_\pi \mathbb{E}\left[\sum_{t \geq 0} \gamma^t r(S_t, g) \,\Big|\, S_0 = s,\, A_0 = a\right], \tag{32}$$

where the supremum is taken over all possible policies. Following the same convention as in the main text, we drop the superscript $\star$ and treat $Q$ as a random variable $Q := Q(S, A, G)$.

We now formalize Q-sufficiency in parallel with our definitions of action sufficiency (Definition 5.2) and value sufficiency (Definition 6.1).

**Definition I.1** (Q-Sufficient Representation). A representation $Z = \phi(S, G)$ is *Q-sufficient* if there exists a measurable function $q$ such that

$$Q(S, A, G) = q(S, A, Z) \quad \text{a.s.} \tag{33}$$

Equivalently, $I(Q; G \mid S, A, Z) = 0$.

Q-sufficiency states that $Z$ preserves all information about the optimal Q-function: once $(S, A, Z)$ is given, $G$ provides no additional information about $Q$.

To relate Q-sufficiency to action-sufficiency, we adopt a canonical specification of the optimal action distribution:

**Assumption I.2** (Uniform Optimal Policy). For each $(s, g)$, the optimal action distribution $P(A \mid S = s, G = g)$ is the uniform distribution over the argmax set

$$\arg\max_{a \in \mathcal{A}} Q(s, a, g). \tag{34}$$

This assumption is consistent with the principle of maximum entropy (Jaynes, 1957): among all optimal policies, which by definition achieve the maximum expected return, the uniform distribution over the argmax set has the highest entropy. We note that this assumption is canonical rather than restrictive, as the same Q-sufficiency argument extends to any tie-breaking rule that depends only on $(s, a, z)$.

**Proposition I.3** (Q-Sufficiency Implies Action-Sufficiency). *Under Assumption I.2, if $Z = \phi(S, G)$ is Q-sufficient, then it is action-sufficient, i.e., $I(A; G \mid S, Z) = 0$.*

*Proof.* By Q-sufficiency, there exists a measurable function $q$ such that $Q(s, a, g) = q(s, a, z)$ for $z = \phi(s, g)$ almost surely. Hence, for any $(s, g)$ mapped to $z$,

$$\arg\max_a Q(s, a, g) = \arg\max_a q(s, a, z), \tag{35}$$

so the argmax set is determined by $(s, z)$ alone.

By Assumption I.2, $P(A \mid S = s, G = g)$ is the uniform distribution over this argmax set. Since the set depends on $(s, g)$ only through $z$, the conditional distribution $P(A \mid S = s, G = g)$ is also determined by $(s, z)$ alone. That is,

$$P(A \mid S, G) = P(A \mid S, Z) \quad \text{a.s.} \tag{36}$$

This implies $A \perp G \mid (S, Z)$, and therefore $I(A; G \mid S, Z) = 0$, i.e., $Z$ is action-sufficient. $\square$

## I.2. Empirical Comparison

We empirically evaluate Q-based goal representations on the OGBench `cube` tasks. The Q-based representation $\phi_Q$ is obtained by training an action-conditioned Q-function $Q_\eta(s, a, \phi_Q(s, g))$ with the encoder $\phi_Q$ jointly learned via the standard TD objective. The encoder $\phi_Q$ is then frozen and used as the goal representation for the hierarchical policy, in direct analogy with how $\phi_V$ and $\phi_A$ are used in the main text.

*Table 6.* Success rate comparison of Q-based and actor-based representations on `cube` datasets. Results are reported as mean $\pm$ std, averaged over 4 seeds.

| Goal Rep. | cube-double-play | cube-triple-play |
|---|---|---|
| Q Rep. ($\phi_Q$) | $0.42 \pm 0.04$ | $0.32 \pm 0.03$ |
| Actor Rep. ($\phi_A$, ours) | $\mathbf{0.42 \pm 0.03}$ | $\mathbf{0.41 \pm 0.04}$ |

Table 6 shows that the Q-based representation achieves performance comparable to the actor-based representation, matching it on `cube-double-play` and trailing modestly on `cube-triple-play`. This is consistent with Proposition I.3: under our assumption, Q-sufficiency implies action-sufficiency, and a representation that is Q-sufficient should yield an effective planner-controller interface.

These empirical results support our broader thesis: the key property of an effective goal representation is action sufficiency, rather than the specific learning objective used to obtain it. Representations trained via different objectives—actor NLL or Q-learning—can both yield action-sufficient features, provided they target the structure necessary for optimal action prediction. This stands in contrast to value-based representations, which optimize for a scalar value signal that may discard action-relevant information within value level sets.

# J. High-Level Predictability of Goal Representations

A goal representation $\phi$ plays a dual role in hierarchical control: it must (*i*) preserve sufficient information for the low-level policy to predict optimal actions (action sufficiency), and (*ii*) be compact and predictable enough for the high-level policy to generate it as a subgoal. While the main text focuses on the former, this appendix empirically examines the latter. We address a natural concern: does pursuing action sufficiency come at the cost of high-level predictability?

## J.1. Motivation

As noted in our discussion on the necessity of goal representations (Section 9) and in our limitations (Section 10), an effective goal representation must serve as an interface between the planner and the controller—preserving action-relevant information while remaining predictable for the high-level policy. Raw goals and autoencoder-based representations, for instance, achieve high action sufficiency but retain nuisance information that makes high-level subgoal prediction harder. A natural concern is whether our actor-based representation $\phi_A$, which explicitly preserves action-relevant information, might itself burden the high-level policy with additional predictive difficulty.

## J.2. Evaluation Protocol

To assess high-level predictability, we measure how accurately the trained high-level policy $\pi^h$ predicts the representation of oracle subgoals along optimal trajectories. Specifically, given an oracle trajectory $\tau^\star = (s_0, s_1, \ldots, s_t, \ldots, s_{t+k}, \ldots, s_T)$, we compute the L2 prediction loss between the high-level policy output and the representation of the corresponding $k$-step oracle subgoal:

$$\mathcal{L}_{\text{pred}}(\phi) = \mathbb{E}_{\tau^\star, t}\left[\left\|\pi^h(s_t, g) - \phi(s_t, s_{t+k})\right\|_2^2\right]. \tag{37}$$

A lower $\mathcal{L}_{\text{pred}}$ indicates that the representation of $k$-step subgoals is more predictable from $(s_t, g)$.

## J.3. Results

Table 7 shows that the actor-based representation $\phi_A$ achieves *lower* high-level prediction loss than the value-based representation $\phi_V$ on both `cube` tasks. This indicates that pursuing action sufficiency does not come at the cost of high-level predictability; if anything, the actor-based representation is more amenable to high-level prediction than its value-based counterpart in our experiments.

*Table 7.* High-level prediction loss $\mathcal{L}_{\text{pred}}$ on `cube` tasks (`play` datasets), averaged over 4 seeds. Lower is better.

| Goal Rep. | cube-double-play | cube-triple-play |
|---|---|---|
| Value Rep. ($\phi_V$) | $0.53 \pm 0.01$ | $0.66 \pm 0.01$ |
| **Actor Rep. ($\phi_A$, ours)** | $\mathbf{0.30 \pm 0.02}$ | $\mathbf{0.45 \pm 0.01}$ |

### J.4. Discussion

This result complements the low-level success rate improvements reported in the main text (Tables 1 and 2). Taken together, the two findings indicate that the actor-based representation simultaneously improves both ends of the hierarchical interface: it strengthens low-level controllability while not harming—and even modestly improving—high-level predictability.

We caution against over-interpreting this finding as a general guarantee. Predictability depends on multiple factors beyond the representation's information content, including the smoothness of the latent space and architectural choices. As we note in our limitations (Section 10), balancing action-relevant information with the compressibility required for efficient high-level planning remains an important direction for future work. Nevertheless, our results demonstrate that action sufficiency and high-level predictability are not inherently in tension, supporting the actor-based representation as a viable planner-controller interface.

