# OpenReview forum: "Action-Sufficient Goal Representations"
_ICML.cc/2026/Conference — ICML 2026 regular_

### Official Review · Reviewer_Wp2z · 2026-02-20

**Soundness:** 4
**Presentation:** 3
**Significance:** 3
**Originality:** 3
**Overall Recommendation:** 5
**Confidence:** 5

**Summary:**

The paper discusses the notion of action-sufficient representations and studies it in the context of hierarchical offline goal-conditioned RL, in particular, the HIQL algorithm. They argue that while HIQL learns goal embeddings as part of value learning, that is not a principled method to learn goal embeddings because essentially two goal states could have the same value function but require different actions to be reached, for instance, two goals that are at the same distance from a state but in different directions. They provide didactic examples to show the problem. They define the conditional KL risk to measure the discrepancy between the optimal policy and the trained policy and show its decomposition into a modelling error and a representation error. They then define the notion of value sufficiency and action sufficiency and theoretically show that a representation can be value sufficient but not action sufficient. They then use the example of a discrete cube to assess different goal representations and show that representations could, in general, be value sufficient but not action sufficient and vice versa, and show that among value sufficient representations, only the ones that are action sufficient as well have a high success rate. They then suggest changing the HIQL algorithm to learn the representations as part of the policy loss, and they show that minimizing the cross-entropy loss corresponds to minimizing the conditional KL risk; therefore, the representations trained using this loss are action sufficient. And finally, they try their method on a family of cube tasks and report a better success rate compared to the baseline that trains representations with the value function.

**Compliance With Llm Reviewing Policy:**

Affirmed.

**Final Justification:**

The paper is solid. I only had concerns about the limited literature review and making overly strong claims about the fact that this whole analysis is only done by them, but the authors promised to make the related work more complete and avoid using strong language in their claims, which completely resolved my concern. The paper topic is very useful to the field, and the authors have done a great job by providing theoretical and empirical evidence for their claim, and they have been very responsive during the rebuttal. I keep the high score of 5 and hope the paper gets in.

**Key Questions For Authors:**

1) The main annoying thing while reading this paper to me was the fact that the authors didn't acknowledge the previous literature on the notion of actionable or action-sufficient representation. They say they challenge the implicit assumption that value function features facilitate policy learning, but this fact has already been studied by previous work. I still find this paper very interesting, and the contributions are significant, but I think it is not fair to the readers to frame the paper as such this problem hasn't been studied before, so I kindly ask the authors to revise their introduction and related work to put more weight on the previous works and contextualize their work in the literature better.

2) Do you need any regularization in the policy network to make sure only the action-relevant features are captured? It will be very helpful if you add a short discussion about that as well, because a representation that captures everything is action sufficient but highly inefficient, so it is nice to know if there is any trick used in practice to make this work efficiently.

**Limitations:**

yes

**Strengths And Weaknesses:**

Soundness: The authors use multiple theoretical and empirical pieces of evidence to verify the correctness of their claims. The results make complete sense to me and are very intuitive. I checked most of the proofs they seem correct.

Presentation: The presentation is fantastic! They do a great job at familiarizing the potentially unfamiliar reader with the problem and providing very intuitive, simple examples to show the severity of the problem. Their discrete cube example is very elegant and confirms their theoretical discussions in a very neat way. The theoretical results are also explained in a very clear way, and in general, I had a fun time reading the paper. My only problem with the presentation is that they put the related work in the appendix, and they neglect to appreciate both in the main text and the appendix that the notion of action-sufficiency is not introduced by them for the first time; this concept is indeed studied in the literature of inverse dynamic models, causal RL, and empowerment. They should also cite the paper "Studying the interplay between the actor and critic representations in reinforcement learning" by Garcin et al., which studies the difference between representations learnt by the actor and critic. Another related work is "LEARNING ACTIONABLE REPRESENTATIONS WITH GOAL-CONDITIONED POLICIES" by Ghosh et al.

Significance: Learning compact and sufficient representations in RL is a very important problem, and coming up with principled methods that incorporate the representation learning in the practical offline RL algorithms is a very important task.

Originality: While the notion of action sufficiency and pinning down the difference between actor and critic representations is not their novel contribution, I think the framing they do in the context of hierarchical offline RL is certainly novel and original. Moreover, by applying their method, they manage to make a significant improvement in HIQL.

---

> ### Author Rebuttal · Authors · 2026-03-31
>
> # Reviewer 4 (Wp2z)
>
> ## Question 1: Missing related work and overstatement of novelty
>
> We thank the reviewer for the constructive feedback. We agree that it would be an overstatement to claim we are the first to propose the necessity of representations sufficient for action prediction. The importance of representation learning has long been emphasized across the broader RL literature, and our study strongly follows this lineage, specifically bringing it into the scope of offline GCRL.
>
> As you pointed out, Ghosh et al. [1]  introduced "actionable representations" that capture only the elements necessary for decision-making, demonstrating their applicability to hierarchical RL. This forms a vital foundation for the action sufficiency we formulated. Furthermore, Garcin et al. [2] theoretically and empirically verified that actor and critic representations specialize in different information, noting the actor's tendency to focus on action-relevant features. This shares our exact philosophy: value-centric representations can be fundamentally ineffective for controllability. We also acknowledge that our work shares its core concept—"representations for action prediction"—with several other prominent studies [3, 4, 5, 6, 7] and omitting them can mislead readers into thinking we were the first to address this concept.
>
> To clarify, our main contribution is not the general idea of action-centric representations. Rather, it lies in **the information-theoretic formalization and empirical verification of action sufficiency specifically as a required property for goal representations serving as the planner-controller interface in hierarchical methods for offline GCRL**.
>
> We will ensure that these updates are fully reflected in the camera-ready manuscript. We will bring the Related Works section into the main text and explicitly incorporate these works to clearly define the relationship between prior literature and our specific contributions, ensuring there is no room for misinterpretation.
>
> ## Question 2: Whether additional regularization is needed
>
> While our proposed actor NLL objective encourages action sufficiency, it does not *explicitly* regularize the representation for retaining action-irrelevant nuisance information.
>
> However, we do introduce this regularization *implicitly* through our architectural design. Specifically, following the standard architectural practice in the original HIQL framework, we employ a bottleneck layer for the goal representation, compressing the raw state/goal vector dimension (which is typically around 30 to 50 in OGBench cube tasks) down to a compact latent space (dim = 10). While dimensional reduction does not strictly guarantee information-theoretic compression, we opted for this dimensional bottleneck as the simplest, most practical solution, and we believe it contributes sufficiently to the necessary compression.
>
> This intuition is empirically supported by our experiments using the original, uncompressed goals, as shown in Table 2 in our manuscript. Theoretically, utilizing the raw goal vector ensures trivial action sufficiency because it drops zero information. However, as you mentioned, it also retains all the unnecessary information, which makes it unsuitable for the high-level policy's planning and prediction phases. Our results in Table 2 reveal that the raw goal is empirically suboptimal compared to our actor representations. We believe this performance gap provides supporting evidence for our postulation: the bottleneck layer could act as an implicit compressor.
>
> While our current bottleneck architecture provides a practical degree of implicit compression, we also note that applying explicit information-theoretic regularizers (*e.g.*, variational information bottleneck [8, 9]) to strictly enforce minimality is a promising avenue for future work.
>
>
> [1] Ghosh et al., Learning Actionable Representations with Goal-Conditioned Policies, ICLR, 2019.
>
> [2] Garcin et al., Studying the Interplay Between the Actor and Critic Representations in Reinforcement Learning, ICLR, 2025.
>
> [3] Huang et al., Action-Sufficient State Representation Learning for Control with Structural Constraints, ICML, 2022.
>
> [4] Lamb et al., Guaranteed Discovery of Control-Endogenous Latent States with Multi-Step Inverse Models, TMLR, 2023.
>
> [5] Islam et al., Principled Offline RL in the Presence of Rich Exogenous Information, ICML, 2023.
>
> [6] Brandfonbrener et al., Inverse Dynamics Pretraining Learns Good Representations for Multitask Imitation, NeurIPS, 2023.
>
> [7] Rudolph et al., Learning Action-based Representations Using Invariance, RLJ, 2024.
>
> [8] Tishby et al., The Information Bottleneck Method, Allerton Conference on Communication, Control, and Computing, 1999.
>
> [9] Alemi et al., Deep Variational Information Bottleneck, ICLR, 2017.

---

> > ### Author Rebuttal · Reviewer_Wp2z · 2026-04-03
> >
> > Thanks to the authors, I trust that they will add the discussed related works to their paper and will keep the high score. Good luck.

---

> > > ### Author Response · Authors · 2026-04-06
> > >
> > > Thank you very much for the encouraging feedback. We sincerely appreciate your thoughtful comments and suggestions, and we will make sure that the discussed related works are clearly incorporated into the revised manuscript.

---

### Official Review · Reviewer_TwQ3 · 2026-03-08

**Soundness:** 3
**Presentation:** 2
**Significance:** 3
**Originality:** 2
**Overall Recommendation:** 4
**Confidence:** 2

**Summary:**

This paper questions a common assumption in hierarchical offline GCRL: goal representations optimized for value learning are necessarily suitable for low-level control. It introduces an information-theoretic notion of action sufficiency, proves that value sufficiency does not imply action sufficiency, and shows via oracle-subgoal evaluations and OGBench cube tasks that actor-derived representations can substantially improve low-level success.

**Compliance With Llm Reviewing Policy:**

Affirmed.

**Final Justification:**

the rebuttal addresses my concerns

**Key Questions For Authors:**

1. Impact on high-level planning: With actor-derived representations, does high-level subgoal modeling and sampling become harder? Can the authors report high-level planning quality (e.g., subgoal predictability, planning success, or high-level policy NLL) to clarify where end-to-end gains come from?

2. Generality beyond cubes: Do the low-level failures of value-centric representations persist under pixel observations or across more diverse tasks beyond cubes? Would performing action-sufficiency-aligned learning in the encoder latent space improve robustness and reduce compute?

**Limitations:**

See weakness

**Strengths And Weaknesses:**

Strengths:

1.  Clear and important insight: The paper highlights that being sufficient for value prediction is not equivalent to being discriminative enough for optimal action selection, targeting a key bottleneck in hierarchical GCRL interfaces.
2. Strong theoretical framing: The conditional KL risk decomposition isolates an irreducible representation term and motivates a precise, testable definition of action sufficiency, clarifying when compression limits optimal control.
3. Well-designed experiment to isolate factors: The oracle-subgoal setup effectively removes high-level planning error, making the low-level failure more convincingly attributable to the goal representation.
4. Practical and simple approach: Tying representation learning directly to the actor objective is lightweight to implement and yields intuitive gains.



Weaknesses:

1. Limited empirical breadth: The evaluation focuses primarily on cube manipulation including discrete or analytically tractable settings. It remains unclear whether the conclusions hold for other long-horizon task types (locomotion, navigation), pixel observations, or stronger partial observability.

2. Insufficient analysis of coupling with high-level planning: While action sufficiency is necessary for the low-level controller, hierarchical systems also require representations to be predictable, plan-able and compressible for the high-level policy. The trade-off, e.g., whether more action-sufficient representations make high-level modeling harder, is not systematically analyzed.

3. Baseline coverage for representation learning is narrow: Beyond value-centric vs actor-derived features, comparisons to broader representation objectives are limited, making it harder to judge whether the proposed approach is the best or only solution.

4. Theory-to-offline gap: Action sufficiency is defined at an information-theoretic level, but offline dataset coverage, distribution shift, and AWR approximation error can affect whether such representations can be learned in practice; more sensitivity analysis would strengthen the claims.

---

> ### Author Rebuttal · Authors · 2026-03-31
>
> # Reviewer 3 (TwQ3)
>
> ## Weakness 1: Limited empirical breadth
>
> We thank the reviewer for this helpful comment. As this important concern regarding the limited scope of our experimental evaluations was shared among reviewers, we have provided a comprehensive, unified response—including new evaluations on four additional environments—in **our reply to Reviewer 1 (R2hn) under Weakness 1**. We apologize for the inconvenience of cross-referencing and kindly ask you to refer to that section for the full details. We believe these additional results will help clarify our empirical claims and address your concern.
>
> ## Weakness 2 & Question 1: Impact on high-level planning and predictability
>
> It is a valid intuition that preserving more action-relevant information could increase the predictive burden on the high-level policy, potentially leading to a trade-off. However, our point is not to "add" extra action information on top of an existing representation, but to learn the planner-controller interface itself in an action-aware way. In this sense, action sufficiency does not necessarily make subgoal prediction harder; it defines the information that the interface should preserve for downstream control.
>
> To address the reviewer's concern, we evaluated whether action-sufficiency-aware representations make high-level subgoal prediction harder in practice. Specifically, given an oracle trajectory $\tau^\star = (s_0, s_1, \ldots, s_{T-1}, s_T = g)$, we compute the L2 loss between the high-level policy output and the representation of the corresponding oracle subgoal, $\phi(s_t, s_{t+k})$, along the trajectory. The results are as follows:
>
> ||`cube-double-play`|`cube-triple-play`|
> |-|-|-|
> |actor rep. ($\phi_A$) (ours)|0.30 ± 0.02|0.45 ± 0.01|
> |value rep. ($\phi_V$)|0.53 ± 0.01|0.66 ± 0.01|
>
> *(Averaged over 4 seeds.)*
>
> This shows that the actor-based representation does not hurt high-level predictability relative to the value-based baseline. In fact, it achieves lower prediction loss on oracle subgoals in our experiments, suggesting that action-sufficiency-aware representations need not make high-level subgoal prediction harder in practice.
>
> Additionally, as shown in the low-level policy evaluation in our manuscript (Figure 1), the actor-based representation yields a significant performance improvement in low-level controllability over the value-based one.
>
> In conclusion, our results suggest that the actor-based representation does not harm high-level predictability, while providing a clear advantage in low-level controllability. We therefore interpret the end-to-end gains of our method as arising primarily from improved low-level controllability.
>
> ## Weakness 3: Narrow baseline coverage for representation learning
>
> We thank the reviewer for this helpful comment. As this concern substantially overlaps with **Reviewer 2 (qMac)’s Weakness 2 & Questions 2 and 3**, we have provided a unified response there, including additional comparisons with non-value-centric representation baselines and a clarification of the scope of our claims. We apologize for the inconvenience of cross-referencing and kindly ask you to refer to that section for the full details. We believe that response will help address your concern.
>
> ## Weakness 4 & Question 2: Theory-to-offline gap and generality of action sufficiency
>
> We agree that there is a gap between our theoretical assumptions (e.g., optimal action distribution, infinite data) and practical offline RL implementations. However, we clarify that **the concept of action sufficiency itself is not strictly confined to the theoretical level**; rather, it defines the fundamental property that a goal representation must ultimately possess to serve as an effective interface in hierarchical GCRL.
>
> We acknowledge that achieving action sufficiency in practice is a separate challenge. Although Theorem 7.2 provides a principled method to learn these representations, it abstracts away practical offline RL challenges such as limited dataset coverage and AWR approximation errors. While analyzing the precise theoretical impact of these practical approximations is an important direction for future research, a deep theoretical treatment of this gap falls outside the scope of this project.
>
> Instead, we demonstrate the practical generality of our approach empirically. As detailed in our earlier response, we expanded our evaluation across a diverse set of environments. Across these varied and non-ideal offline datasets, the actor-based representation performs comparably to or better than the other baselines. These results indicate that even under practical approximations, pursuing action sufficiency yields robust empirical gains.

---

> > ### Author Rebuttal · Reviewer_TwQ3 · 2026-04-03
> >
> > The rebuttal partially addresses my concerns. In particular, the new analysis on high-level subgoal prediction makes the coupling between action-sufficient representations and high-level planning much clearer, and it is reassuring that the actor-based representation does not appear to hurt predictability in the reported experiments. However, some concerns remain only partially resolved: the broader empirical generality and baseline coverage are addressed mainly via cross-referenced responses, which makes them harder to assess from the current rebuttal alone, and the theory-to-offline gap is acknowledged more than it is resolved. Overall, my concerns are reduced, but not fully resolved.

---

> > > ### Author Response · Authors · 2026-04-06
> > >
> > > We are glad that the additional analysis on high-level policy evaluation clarified the relationship between action-sufficient representations and high-level planning.
> > >
> > > We also appreciate your remaining concerns, and we apologize for the inconvenience caused by cross-referencing. We distributed some of the results across responses due to the character limit. We restate the results here.
> > >
> > > ## Limited empirical breadth
> > >
> > > We expanded the evaluation beyond the original cube tasks to four additional OGBench environments: `humanoidmaze-giant-navigate`, `scene-play`, `visual-cube-double-play`, and `visual-scene-play`. The success rates are:
> > >
> > > ||`humanoidmaze-giant`|`scene`|`visual-cube`|`visual-scene`|
> > > |-|-|-|-|-|
> > > |value rep. ($\phi_V$)|0.97 ± 0.01|0.65 ± 0.03|0.06 ± 0.02|0.13 ± 0.02|
> > > |actor rep. ($\phi_A$) (ours)|0.91 ± 0.01|0.76 ± 0.04|0.55 ± 0.05|0.55 ± 0.02|
> > >
> > > *(Averaged over 3 seeds for the visual-domain tasks and 4 seeds for the remaining tasks.)*
> > >
> > > On `humanoidmaze-giant`, both representations perform comparably well, suggesting value-based representations can sometimes suffice for downstream control. **This does not contradict our claim**: value-based representations are not always inadequate, but since they are not explicitly aligned with action prediction, they can lose action-relevant information and exhibit specific failure modes.
> > >
> > > However, a substantial performance gap between the two representations emerges in **visual domains with pixel-valued state spaces**. We view these results as strong evidence that the **importance of preserving action-relevant information becomes especially visible when nuisance information is abundant**.
> > >
> > > ## Baseline coverage for representation learning
> > >
> > > We additionally evaluated two non-value-centric representation baselines on the cube tasks:
> > >
> > > - **Q-function-based representation**: we train an action-conditioned Q-function $Q(s,a,\phi_Q(s,g))$ and use its encoder $\phi_Q(s,g)$ as the representation.
> > > - **Autoencoding representation**: we jointly train an encoder that compresses $(s,g)$ into a low-dimensional latent $z$, and a decoder that reconstructs $g$ from $(s,z)$, and use $z$ as the representation.
> > >
> > > The success rates are:
> > >
> > > ||`cube-double-play`|`cube-triple-play`|
> > > |-|-|-|
> > > |actor rep. ($\phi_A$) (ours)|0.42 ± 0.03|0.41 ± 0.04|
> > > |Q rep. $(\phi_Q)$|0.42 ± 0.04|0.32 ± 0.03|
> > > |autoencoder rep.|0.35 ± 0.02|0.24 ± 0.04|
> > >
> > > *(Averaged over 4 seeds.)*
> > >
> > > Notably, the **Q-function-based representation** performs comparably to the actor-based one, **which aligns with our thesis**. Assuming the target action distribution is uniform over the argmax set of the optimal Q-function $Q^*$, this connection becomes clear. Concretely, if $Z=\phi_Q(S,G)$ is sufficient for the optimal action-conditioned Q function, then there exists a function $q$ such that:
> > >
> > > $$
> > > Q^*(S,A,G) = q(S,A,Z).
> > > $$
> > >
> > > It follows that, for any $(s,g)$ mapped to $z$,
> > >
> > > $$
> > > \arg\max_a Q^*(s,a,g) = \arg\max_a q(s,a,z),
> > > $$
> > >
> > > so the support of the optimal action distribution is fully determined by $(s,z)$. Since, by assumption, the target action distribution is the uniform distribution over this support, $P(A\mid S,G)$ is also determined by $(S,Z)$ alone. Therefore, $Z$ is action-sufficient.
> > >
> > > The **autoencoding representation** provides a complementary perspective. While it may retain action-relevant information in principle, it also preserves nuisance information, which can hurt high-level predictability and is not always better in hierarchical control.
> > >
> > > ## Theory-to-offline gap
> > >
> > > We would like to address this issue by separating two distinct aspects.
> > >
> > > **Regarding the methodology for learning action-sufficient representations:** we do not claim that this practical gap is fully resolved. Our Theorem 7.2 provides a principled learning objective under ideal assumptions, but our current theoretical analysis does not cover practical offline RL complexities such as data coverage. However, **the empirical evidence in the paper, together with the expanded evaluations discussed above, suggests that the resulting method remains robust and effective in practice even under these non-ideal conditions**.
> > >
> > > **Regarding the concept of action sufficiency itself:** we do not view it as merely a theoretical artifact. Rather, **it identifies the fundamental property that a representation should satisfy in order to serve as an effective planner-controller interface**. In this sense, action sufficiency is the target property we ultimately seek, while the practical difficulties of offline learning should be understood as challenges in approximating that target in realistic settings, rather than as flaws in the concept itself.
> > >
> > > For these reasons, we believe our paper contributes at both levels: **it theoretically formalizes the target property itself, and it provides empirical evidence that pursuing this property can yield robust gains in practice.**
> > >
> > > Again, thank you for your comments and we hope these detailed discussions address your remaining concerns.

---

### Official Review · Reviewer_qMac · 2026-03-12

**Soundness:** 2
**Presentation:** 2
**Significance:** 2
**Originality:** 3
**Overall Recommendation:** 4
**Confidence:** 3

**Summary:**

This paper studies goal representations in hierarchical offline GCRL. The authors show that value-based representations (as in HIQL) can lose action-relevant information even when value estimation is accurate. They formalize this via "action sufficiency," prove value sufficiency does not imply it, and propose learning representations jointly with the actor via NLL minimization. Experiments on a Discrete Cube environment and OGBench cube tasks support the claim.

**Compliance With Llm Reviewing Policy:**

Affirmed.

**Final Justification:**

The rebuttal partially addressed my concerns: the expanded experiments and baselines strengthened the empirical evaluation, leading me to raise my score from 3 to 4. However, my concern about the novelty of the core contribution remains, as the idea of preserving action-relevant information in representations is not new. I encourage the authors to position their contribution more carefully in the camera-ready.

**Key Questions For Authors:**

1. Have you evaluated outside OGBench cube tasks to show that the observed failure mode is not benchmark-specific?
---
2. Is the identified failure mode intended as a critique of HIQL-style value-based hierarchical representations specifically, or of offline GCRL goal representations more broadly?
---
3. If the intended claim is broader than HIQL-style value-based hierarchies, can the authors compare to or discuss non-value-centric approaches such as action-conditioned Q-function methods?

**Limitations:**

Yes, adequately discussed, though the lack of diverse experiments is a more significant limitation than acknowledged.

**Strengths And Weaknesses:**

S1. The 1D thought experiment (Section 3.2) and oracle subgoal evaluation (Section 3.1) effectively motivate the problem.

---

S2. The Discrete Cube enables exact information-theoretic analysis; the (ΔV, ΔA) exploration (Figure 3) cleanly demonstrates the decoupling of value and action sufficiency.

---

W1. Some key presentation choices make the paper harder to follow than necessary. The manuscript assumes substantial prior familiarity with the cube tasks, and some important components, especially AWR, are introduced too late or with too little explanation in the main text.

---

W2. The paper’s framing is broader than its evidence. The main empirical and methodological focus is on HIQL-style hierarchical methods that learn a goal representation through value optimization, so the paper more clearly establishes a limitation of this specific value-representation pipeline than of goal representation learning in offline GCRL more generally. It would strengthen the paper to either compare against at least one non-value-centric baseline, such as a method based on action-conditioned Q-functions, or to narrow the scope of the claim accordingly.

---

W3. The empirical scope is narrow relative to the paper’s broader positioning. The experiments focus almost entirely on OGBench cube tasks. It remains unclear how broadly the conclusions transfer to other environments.

---

> ### Author Rebuttal · Authors · 2026-03-31
>
> # Reviewer 2 (qMac)
>
> ## Weakness 1: Presentation clarity issues
>
> We agree that our manuscript assumes too much familiarity with some specific tasks and introduces key components later than ideal. In the camera-ready version, we will improve the presentation by (1) providing a more self-contained description of the experimental environments, and (2) introducing AWR and other key components earlier in the main text with clearer motivation. We believe these revisions will significantly improve readability.
>
> ## Weakness 2 & Questions 2 and 3: Scope of claims and comparison with non-value-centric representation baselines
>
> > **Shared response to Reviewers 1 (R2hn), 2 (qMac), and 3 (TwQ3).**
>
> We agree that the strongest direct empirical evidence in our current paper is about HIQL-style methods that learn goal representations through state-value optimization.
>
> Indeed, our main critique is aimed at state-value-based representations, which can be too weak to preserve the action-relevant information needed for low-level control, despite being widely adopted in HIQL-style hierarchical baselines. Nonetheless, our intention is not to tie the notion of action sufficiency only to this specific pipeline. Rather, we view action sufficiency as **a broader conceptual lens for asking what information a goal representation must preserve in order to support low-level control**.
>
> To make this distinction more concrete, we followed the reviewer’s suggestion and conducted additional experiments with non-value-centric representation baselines. These experiments are intended to clarify that action sufficiency is not tied only to HIQL-style state-value-based representations, but can also serve as a broader lens for understanding other representation choices.
>
> Specifically, on the cube environments, we evaluated the following additional representations:
>
> - **Q-function-based representation**: We train an action-conditioned Q-function $Q(s,a,\phi_Q(s,g))$ and use its encoder $\phi_Q(s,g)$ as the representation
> - **Autoencoding representation**: We jointly train an encoder that compresses $(s,g)$ into a low-dimensional latent $z$, and a decoder that reconstructs $g$ from $(s,z)$, and use $z$ as the representation
>
> The success rates are as follows:
>
> ||`cube-double-play`|`cube-triple-play`|
> |-|-|-|
> |actor rep. ($\phi_A$) (ours)|0.42 ± 0.03|0.41 ± 0.04|
> |Q rep. $(\phi_Q)$|0.42 ± 0.04|0.32 ± 0.03|
> |autoencoder rep.|0.35 ± 0.02|0.24 ± 0.04|
>
> *(Averaged over 4 seeds.)*
>
> One notable result is that the **Q-based representation** performs comparably to the actor-based one. We view this as consistent with our overall thesis. The connection between Q sufficiecy and action sufficiency can be stated as follows. The optimal action distribution is supported on the argmax set of the optimal Q-function $Q^*$. If we further adopt a canonical target action distribution that is uniform over this support, then Q sufficiency implies action sufficiency. Concretely, if $Z=\phi_Q(S,G)$ is sufficient for the optimal action-conditioned Q function, then there exists a function $q$ such that:
>
> $$
> Q^*(S,A,G) = q(S,A,Z).
> $$
>
> It follows that, for any $(s,g)$ mapped to $z$,
>
> $$
> \arg\max_a Q^*(s,a,g) = \arg\max_a q(s,a,z),
> $$
>
> so the support of the optimal action distribution is fully determined by $(s,z)$. Since, by assumption, the target action distribution is the uniform distribution over this support, $P(A\mid S,G)$ is also determined by $(S,Z)$ alone. Therefore, $Z$ is action-sufficient.
>
> The **autoencoding representation** provides a complementary perspective. If a representation preserves the full goal information, then it need not lose action-relevant information. However, preserving all information, including nuisance information, may reduce predictability for the high-level policy and make the representation less effective as an interface between planning and control. We view this baseline as useful for illustrating that preserving more information is not always better in hierarchical control.
>
> Taken together, these results suggest that the key issue is not whether a representation is actor-based or critic-based per se, but whether it preserves the information needed for downstream control. We therefore view action sufficiency not as a claim tied only to HIQL-style value-based pipelines, but as a general lens for understanding what makes a goal representation effective for hierarchical control.
>
> ## Weakness 3 & Question 1: Limited scope of experimental evaluations
>
> As this important concern regarding the limited scope of our experimental evaluations was shared among reviewers, we have provided a comprehensive, unified response—including new evaluations on four additional environments—in **our reply to Reviewer 1 (R2hn) under Weakness 1**. We apologize for the inconvenience of cross-referencing and kindly ask you to refer to that section for the full details. We believe these additional results will help clarify our empirical claims and address your concern.

---

> > ### Author Rebuttal · Reviewer_qMac · 2026-04-02
> >
> > I thank the authors for the thorough rebuttal. The expanded experiments and additional baselines are encouraging, and I am raising my score from 3 to 4.
> >
> > However, my concern about novelty remains. The core idea that representations should preserve action-relevant information is not new (also noted by Reviewer Wp2z), and I encourage the authors to position the contribution more carefully in the camera-ready.

---

> > > ### Author Response · Authors · 2026-04-06
> > >
> > > We thank the reviewer for recognizing the value of our expanded experiments and additional baselines. We sincerely appreciate your constructive feedback throughout this process.
> > >
> > > Regarding your remaining concern about novelty, we do not claim to be the first to propose the general intuition that representations should preserve action-centric information, as also discussed in our response to Reviewer 4 (Wp2z). We acknowledge that this broader concept has been explored in prior RL literature.
> > >
> > > While we build upon the intuition of action-centric representations, we argue that our core contribution is **the information-theoretic formalization and empirical verification of action sufficiency as a required property for the planner-controller interface in hierarchical offline GCRL**. This perspective not only explains the failure modes of existing state-value-based representations, but also establishes a broader, principled criterion that effective goal representations must satisfy.
> > >
> > > We agree that this point should be stated more precisely, and we will revise accordingly. We will explicitly cite and discuss the related prior works in the main text, ensuring that our specific scope and contributions are distinguished from the broader literature.
> > >
> > > Once again, we sincerely thank you for your insightful comments and for helping us refine the clarity of our work.

---

### Official Review · Reviewer_R2hn · 2026-03-12

**Soundness:** 3
**Presentation:** 3
**Significance:** 3
**Originality:** 3
**Overall Recommendation:** 4
**Confidence:** 2

**Summary:**

This paper studies goal representations in hierarchical offline goal-conditioned reinforcement learning. It introduces action sufficiency, an information-theoretic condition that characterizes whether a representation preserves all goal information necessary for low-level action selection, and shows that value sufficiency does not imply action sufficiency. The paper further argues that representations learned jointly with the actor objective are more likely to be action-sufficient. Experiments in a tractable discrete environment and on OGBench manipulation tasks support the main claim. Overall, the paper addresses a clear problem, presents a well-motivated theoretical perspective, and provides empirical results that are broadly consistent with its central argument.

**Compliance With Llm Reviewing Policy:**

Affirmed.

**Key Questions For Authors:**

Please refer to the weakness.

**Limitations:**

Yes

**Strengths And Weaknesses:**

Strengths:
1. This paper identifies a key issue in hierarchical offline GCRL: whether goal representations should primarily support value estimation or action selection. This is directly relevant to HIQL-style methods.
2. The theoretical story is clear and well-motivated. The notion of action sufficiency is not introduced heuristically, but arises naturally from a conditional KL risk decomposition as the representation error term.

Weaknesses:
1. My main concern is that the range of experimental environments is rather limited, only including the three environments of the OGbench cube. Could the authors expand the scope of the experiments? The author also acknowledged the gap between theoretical analysis and practical application in the "limitation" section. Therefore, a well-designed experiment is essential for verifying the effectiveness of the method.
2. This paper has a relatively small number of baselines in the experimental section and lacks comparisons with other offline goal-conditioned reinforcement learning methods.

---

> ### Author Rebuttal · Authors · 2026-03-31
>
> # Reviewer 1 (R2hn)
>
> ## Weakness 1: Limited range of experimental environments
>
> > **Shared response to Reviewers 1 (R2hn), 2 (qMac), and 3 (TwQ3).**
>
> We thank the reviewer for this valuable feedback. A recurring concern across the reviews was that the current draft has limited empirical evidence. We agree that the empirical scope of the current draft is narrow, and that broader evaluation is important for assessing the practical relevance of our claims.
>
> To address this, we expanded the experiments beyond the original cube tasks. Specifically, we conducted additional experiments on the following OGBench environments:
>
> - `humanoidmaze-giant-navigate`: a maze-based **locomotion and navigation task**
> - `scene-play`: a **robot manipulation task** involving behaviors such as drawer opening and button pressing
> - `visual-cube-double-play`: a **pixel-observation version** of the cube task
> - `visual-scene-play`: a **pixel-observation version** of the scene task
>
> While we acknowledge the existence of other offline GCRL benchmarks, we chose to expand within OGBench because its shared benchmark structure allows us to vary task diversity while keeping the training and evaluation pipeline consistent, making the effect of representation choice easier to interpret. The success rates are as follows:
>
> ||`humanoidmaze-giant`|`scene`|`visual-cube`|`visual-scene`|
> |-|-|-|-|-|
> |value rep. ($\phi_V$)|0.97 ± 0.01|0.65 ± 0.03|0.06 ± 0.02|0.13 ± 0.02|
> |actor rep. ($\phi_A$) (ours)|0.91 ± 0.01|0.76 ± 0.04|0.55 ± 0.05|0.55 ± 0.02|
>
> *(Averaged over 3 seeds for the visual-domain tasks and 4 seeds for the remaining tasks.)*
>
> On `humanoidmaze-giant-navigate`, both value-based and actor-based representations achieve high performance. We do not view this as contradicting our claim. Rather, it is consistent with the possibility that, in some settings, the information needed for value prediction substantially overlaps with the information needed for action selection, so a state-value-based representation can still perform well in practice.
>
> On `scene-play`, the actor-based representation performs somewhat better, but the gap remains modest, suggesting that the value-based representation still preserves a meaningful amount of action-relevant information.
>
> In contrast, the gap becomes much larger in the visual domains. On `visual-cube-double-play` and `visual-scene-play`, the value-based representation degrades sharply, while the actor-based representation remains substantially stronger. This is consistent with our central intuition: in visual environments, representation learning must discard nuisance information while retaining the information required for control, and failures in this process are more clearly exposed.
>
> We believe these expanded evaluations substantially strengthen the empirical foundation of our claims. We will incorporate these new environments, along with the corresponding analyses, into the camera-ready version of the manuscript to provide a more comprehensive assessment of our proposed framework.
>
> ## Weakness 2: Limited comparison with other offline GCRL baselines
>
> We thank the reviewer for this helpful comment. Since our goal is not to propose a fundamentally new offline GCRL algorithm, but rather to understand what makes a goal representation effective for low-level control, we addressed this concern by expanding our comparison to additional representation baselines. As this point substantially overlaps with **Reviewer 2 (qMac)’s Weakness 2 & Questions 2 and 3**, we provide a unified response there, including the added non-value-centric baselines and the corresponding discussion. We apologize for the inconvenience of cross-referencing and kindly ask the reviewer to refer to that section for the full details.

---

> > ### Author Rebuttal · Reviewer_R2hn · 2026-04-01
> >
> > I thank the authors for incorporating additional tasks from OGBench, which address some of my concerns. Regarding the baselines, the authors have added comparisons with Q-function and autoencoder-based methods. I believe that including further existing works as baselines (e.g., Physics-informed Value Learner for Offline Goal-Conditioned RL and Generative Trajectory Stitching through Diffusion Composition) would help the community better understand the significance of this work. However, I understand that the focus of the paper is on goal representations; therefore, my overall score is positive.

---

> > > ### Author Response · Authors · 2026-04-06
> > >
> > > We sincerely thank the reviewer for the continued constructive discussion and for the positive overall assessment. We are very glad to hear that the additional experiments on the OGBench environments helped address your concerns.
> > >
> > > Regarding the two interesting works you mentioned, we agree that they share significant common ground with our paper under the broader umbrella of offline GCRL.
> > >
> > > **Physics-informed Value Learner for Offline Goal-Conditioned RL** proposes a more systematic methodology to learn the goal-conditioned value function (GCVF) via physics-informed learning objectives. However, because its primary focus is to improve the estimation of a *state-value* function, we view it as somewhat orthogonal to our approach. As our theoretical analysis and the empirical comparison against the HIQL baselines demonstrate, state-value functions—even when learned accurately—do not in general guarantee preservation of the action-relevant information necessary for low-level control. Therefore, we believe that the fundamental representation bottleneck we highlight might still apply to this method.
> > >
> > > **Generative Trajectory Stitching through Diffusion Composition**, on the other hand, follows the lineage of diffusion-based planning (*e.g.*, Diffuser [1]). Because it generates trajectories directly rather than utilizing a hierarchical planner-controller architecture that requires a distinct latent goal representation interface, a direct empirical comparison is less straightforward within the specific scope of our representation-focused evaluation.
> > >
> > > Nevertheless, we completely agree with your insight that understanding how these different approaches can be compared is highly relevant to the community and remains an important direction for future research.
> > >
> > > While a direct empirical comparison falls outside the main focus of this paper, your point is valid. We will incorporate a discussion of both works in our Related Works section to provide readers with a more comprehensive view of the offline GCRL community.
> > >
> > > Thank you again for your valuable feedback and for your positive evaluation of our work.
> > >
> > > [1] Janner et al., Planning with Diffusion for Flexible Behavior Synthesis, ICML, 2022.

---

### Decision · Program_Chairs · 2026-04-30

**Decision:**

Accept (regular)

**Comment:**

This paper introduces action sufficiency as a necessary criterion for goal representations in hierarchical offline GCRL, and proves that value-sufficient representations can still be insufficient for low-level control. It motivates the issue with risk decomposition and controlled cube/oracle-subgoal analyses, and shows that learning representations via the actor log-loss improves controllability compared to value-derived embeddings.

Reviewers are broadly positive (mostly weak accept, one accept). Main concerns were limited empirical breadth/baselines and novelty/positioning. The rebuttal strengthens the evidence with additional OGBench tasks (including visual domains where value-based reps degrade markedly), added Q-based/autoencoder baselines, and analysis suggesting actor-based reps do not harm high-level predictability.

Remaining request is to tighten claims and properly contextualize related work on actionable/action-sufficient representations.

Recommendation: Weak accept (leaning accept), conditional on clearer positioning/related work and integrating the added experiments/baselines into the camera-ready.